# Dependency of simulated tropical Atlantic current variability on the wind forcing

Kristin Burmeister[1], Franziska U. Schwarzkopf[2], Willi Rath[2], Arne Biastoch[2,3], Peter Brandt[2,3], Joke F. Lübbecke[2], and Mark Inall[1,4]

[1]SAMS Scottish Association for Marine Science, Oban, United Kingdom
[2]GEOMAR Helmholtz Centre for Ocean Research Kiel, Kiel, Germany
[3]Christian-Albrechts-Universitaet, Kiel, Germany
[4]University of the Highlands and Islands, United Kingdom

**Correspondence:** Kristin Burmeister (Kristin.Burmeister@sams.ac.uk)

**Abstract.** The upper wind-driven circulation in the tropical Atlantic Ocean plays a key role in the basin wide distribution of water mass properties and affects the transport of heat, freshwater, and biogeochemical tracers such as oxygen or nutrients. It is crucial to improve our understanding of its long-term behaviour which largely relies on model simulations and the applied forcing due to sparse observational data coverage especially before the mid-2000s. Here, we apply two different forcing products, the Coordinated Ocean-Ice Reference Experiments (CORE) v2 and the JRA55-do surface dataset, to a high-resolution ocean model. Where possible, we compare the simulated results to long-term observations. We find large discrepancies between the two simulations regarding the wind and current field. In the CORE simulation strong, large-scale wind stress curl amplitudes above the upwelling regions of the eastern tropical North Atlantic seem to cause an overestimation of the mean and seasonal variability of the eastward subsurface current just north of the equator. The wind stress curl of the JRA55-do forcing shows much finer structures and the JRA55-do simulation is in better agreement with the mean and intraseasonal fluctuations of the subsurface current found in observations. The northern branch of the South Equatorial Current flows westward at the surface just north of the equator. On interannual to decadal time scales, it shows a high correlation of R=0.9 with the zonal wind stress in the CORE simulation, but only a weak correlation of R=0.35 in the JRA55-do simulation. We also identify similarities between the two simulations. The strength of the eastward flowing North Equatorial Countercurrent located between 3°N and 10°N co-varies with the strength of the meridional wind stress just north of the equator on interannual to decadal time scales in the two simulations. Both simulations present a comparable mean, seasonal cycle and trend of the eastward off-equatorial subsurface current south of the equator but underestimate the current strength by half compared to observations. In both simulations the eastward flowing Equatorial Undercurrent weakened between 1990 and 2009. In the JRA simulation, which covers the modern period of observations, the Equatorial Undercurrent strengthened again between 2008 to 2018 which agrees with observations, albeit the simulation underestimates the strengthening by over a third. We propose that long-term observations, once they have reached a critical length, need to be used to test the quality of wind-driven simulations. This study presents one step in this direction.

# 1 Introduction

The tropical Atlantic circulation plays a crucial role in the distribution of heat, freshwater, carbon and ecosystem relevant quantities in the Atlantic Ocean. A unique feature of the Atlantic Ocean is the Atlantic Meridional Overturning Circulation (AMOC). The return flow of the AMOC in the upper ocean transports heat, freshwater and biogeochemical properties like carbon or oxygen northward through the basin impacting climate and ecosystems in the entire Atlantic sector. On their way through the tropics water masses experience an important transformation gaining heat (0.22 PW, Hazeleger and Drijfhout, 2006) and salinity (freshwater divergence of 0.16 Sv, Hazeleger and Drijfhout, 2006). About one third of the northward flow is recirculated within the tropical Atlantic current system (Hazeleger and Drijfhout, 2006; Tuchen et al., 2022a). While observations allow now to describe the mean to sub-decadal variability of the upper tropical Atlantic circulation (e.g. Tuchen et al., 2022a; Brandt et al., 2021; Burmeister et al., 2020), the study of decadal changes and trends largely relies on model output (e.g. Burmeister et al., 2019; Hüttl-Kabus and Böning, 2008; Duteil et al., 2014).

The flow field in the tropical Atlantic represents a superposition of shallow meridional overturning cells, the horizontal wind-driven gyre circulation and the basin-wide AMOC (e.g. Schott et al., 2004; Hazeleger and Drijfhout, 2006; Perez et al., 2014; Tuchen et al., 2022a; Heukamp et al., 2022). The currents are thus a result of the easterly trade winds and the resultant equatorial Ekman divergence, the wind stress curl fields in the tropics and subtropics as well as buoyancy and wind forcing at higher latitudes. In the upper ∼300 m, the shallow subtropical cells (STCs) consist of poleward Ekman transport at the surface and equatorward transport in the thermocline which connect the subduction regimes in the subtropics and the upwelling regimes in the tropics (Schott et al., 2004). Upwelling in the tropical Atlantic occurs along the equator east of about 20°W, within the Guinea and Angola Domes, and within the eastern boundary upwelling systems off the coast of West Africa. The strength of the STCs is related to the equatorial Ekman divergence (Tuchen et al., 2019; Rabe et al., 2008) and can impact the strength of the zonal currents in the tropical Atlantic, especially the Equatorial Undercurrent (EUC; Rabe et al., 2008; Brandt et al., 2021). The tropical overturning cells (TCs) are part of the STCs and dominate the meridional flow field in the upper 100 m between 5°N and 5°S (e.g. McCreary and Lu, 1994; Schott et al., 2004; Molinari et al., 2003; Perez et al., 2014). They are governed by wind-driven equatorial upwelling, poleward Ekman transport in the upper limb, off-equatorial down-welling at about ±3 − 5° latitude and a geostrophic flow directed equatorward in the lower limb (e.g. Perez et al., 2014). The shallowest overturning cell is the Equatorial Roll in the upper 80 m along the equator. The southerly wind stress at the equator drives its northward cross-equatorial flow near the surface and southward flow below (Heukamp et al., 2022). A complex system of alternating eastward and westward narrow current bands and strong western boundary currents with northward flow participates in or is superimposed on the STCs, TCs and the Equatorial Roll (e.g. Schott et al., 2004).

The wind-driven gyre circulation in the tropical Atlantic can be largely explained by Sverdrup dynamics: that is the relationship between wind stress curl and depth integrated meridional transport. The trade winds converge in the Intertropical Convergence Zone (ITCZ) slightly north of the equator. The weakening of the north- and south-easterly trade winds towards the ITCZ is associated with a positive and negative wind stress curl north and south of the ITCZ, respectively. According to the Sverdrup dynamics this results in two wind-driven gyres, the tropical gyre north and the equatorial gyre south of the ITCZ

(e.g., Fratantoni et al., 2000). Below the ITCZ, an eastward flowing geostrophic current exists between 3° and 13°N, the North Equatorial Counter Current (Fig. 1; Urbano et al., 2006). In the north and in the south, it is flanked by the westward flowing North Equatorial Current (NEC) and South Equatorial Current (SEC), respectively. Associated with the northward displace-
60 ment of the ITCZ, the SEC reaches into the northern hemisphere and literature often distinguishes between a southern branch (sSEC; south of 10°S), a central branch (cSEC; south of the equator) and a northern branch (nSEC; centred at about 2°N) (e.g. Peterson and Stramma, 1991; Schott et al., 2004).

Persistent easterly winds along the equator push the surface waters towards the west causing the thermocline to slope upwards to the east and hence driving, amongst other factors, the eastward flowing subsurface EUC along the equator (Pedlosky, 1987;
Wacongne, 1989). The EUC supplies water masses from the western basin mostly of southern subtropical origin towards the central and eastern upwelling regions (e.g. Bourlès et al., 2002; Schott et al., 2004; Brandt et al., 2006). Two off-equatorial eastward flowing subsurface currents exist in the Atlantic, the North Equatorial Undercurrent (NEUC) and the South Equatorial Undercurrent (SEUC) centred at about 5°N/S, respectively. Potential driving mechanisms of the NEUC and SEUC are still not fully understood. Assene et al. (2020) investigated the formation and maintenance of the off-equatorial subsurface currents in
the Gulf of Guinea and highlighted the link between submesoscale processes, mesoscale vortices and mean currents which can include any of the driving mechanisms suggested in previous studies: eddy fluxes (Jochum and Malanotte-Rizzoli, 2004), meridional advection (Wang, 2005; Johnson and Moore, 1997; Marin et al., 2000; Hua et al., 2003; Marin et al., 2003; Ishida et al., 2005), lateral diffusion of vorticity (McPhaden, 1984) and the pull by upwelling in the eastern basin (McCreary et al., 2002; Furue et al., 2007, 2009). Please note that some of these studies focus on the Pacific counterparts of the NEUC and
SEUC; due to the resemblance of the equatorial Atlantic and Pacific zonal current structure (e.g. Schott et al., 2004) processes observed in the Pacific off-equatorial undercurrents are thought to apply also in the Atlantic (Assene et al., 2020).

The zonal currents in the tropical Atlantic (Fig. 1) form an interhemispheric buffer for the AMOC. A quantification of the different AMOC pathways in the tropical Atlantic was done by Tuchen et al. (2022a). The main part of the upper AMOC limb enters the tropical Atlantic within the westward flowing sSEC that bifurcates into the northward flowing North Brazil
Undercurrent (NBUC) and the southward flowing Brazil Current at about 15°S. The NBUC merges with the cSEC north of about 5°S and the northward western boundary current becomes a surface intensified current and is called North Brazil Current (NBC). Within the NBC the AMOC finally crosses the equator (e.g. Schott et al., 2004; Hazeleger and Drijfhout, 2006; Rühs et al., 2015). After overshooting the equator, the NBC partly retroflects into the zonal current field, partly breaks up into northward propagating NBC rings (Johns et al., 2003). The EUC, NEUC and NECC feed on the retroflection of the
NBC (Bourlès et al., 1999; Hüttl-Kabus and Böning, 2008; Rosell-Fieschi et al., 2015; Stramma et al., 2005). Furthermore, the NEUC and NECC are partly supplied by water masses of northern hemisphere origin from the retroflection of the westward flowing North Equatorial Current which is part of the subtropical gyre in the North Atlantic (e.g. Schott et al., 1998; Bourlès et al., 1999; Urbano et al., 2008). The eastward currents connect the subducted water masses from the subtropical gyres with the central and eastern upwelling regions in the tropical Atlantic thereby ventilating the oxygen poor eastern basin (Stramma
et al., 2008; Urbano et al., 2008; Hahn et al., 2014; Brandt et al., 2015; Hahn et al., 2017; Burmeister et al., 2019, 2020).

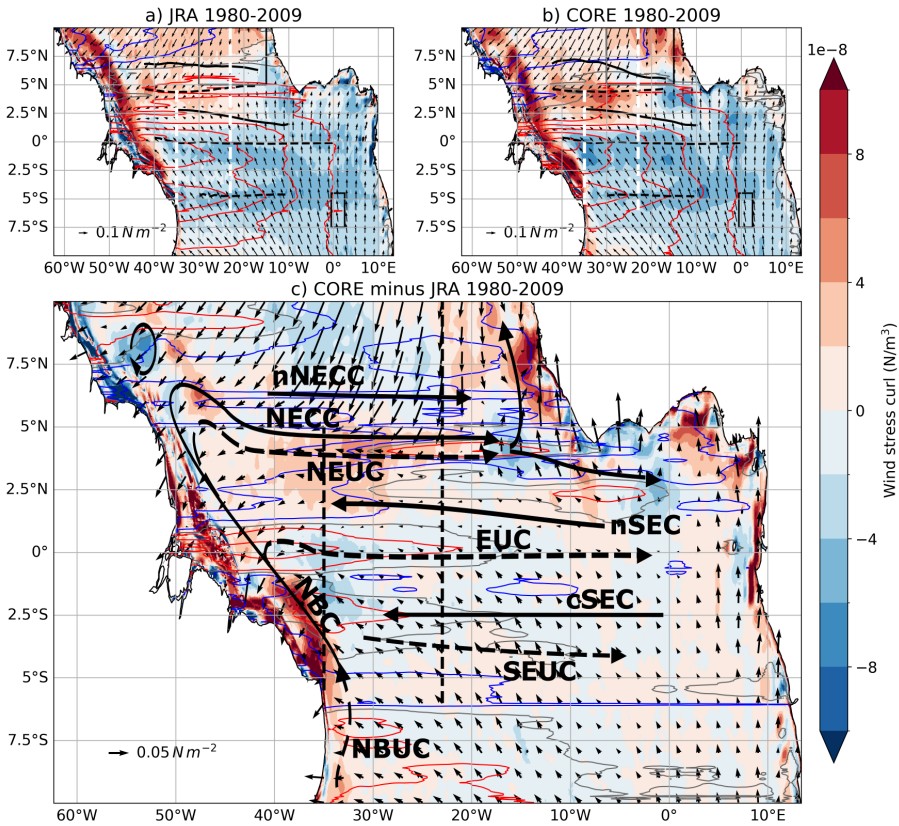

**Figure 1.** (a-c) 1980 to 2009 mean maps of wind stress (black arrows), wind stress curl (colour shading) and associated Sverdrup stream function (contour lines) calculated using (a) $JRA_{sim}$, (b) $CORE_{sim}$, (c) difference between the two forcings. Blue contour lines show negative, red contour lines positive values of Sverdrup stream function, zero line marked as grey contour. A negative stream function presents an anticlockwise rotation, this means that a zero-contour of the stream function with negative values in the south (north) marks maximum westward (eastward) velocities. In (a-b) the contour line interval is 2 Sv, in (c) the $\pm 1.5$ Sv, $\pm 0.5$ Sv isolines are shown. Zonal black lines in (a,b) mark the mean latitude ($Y_{CM}$, Equ. 1) of the simulated surface (solid) and subsurface (dashed) currents for the respective periods, meridional white dashed lines in (a-b) and black dashed lines in (c) mark 35°W and 23°W sections. The black rectangles in (a,b) mark the upwelling regions of the Guinea Dome in the northern and the Angola Dome in the southern hemisphere. (c) Superimposed in black are surface (solid) and thermocline (dashed) currents (adapted from Burmeister et al., 2019, based on observations): North Equatorial Countercurrent (NECC), northern branch of the NECC (nNECC), North Equatorial Undercurrent (NEUC), northern branch of the South Equatorial Current (nSEC) and central branch of the South Equatorial Current (cSEC), Equatorial Undercurrent (EUC), South Equatorial Undercurrent (SEUC), North Brazil Undercurrent (NBUC) and North Brazil Current (NBC).

In the equatorial Atlantic, the enhanced semi-annual to interannual variability of the zonal flow can be attributed to basin resonances of the gravest basin mode (Thierry et al., 2004; Ascani et al., 2006; d'Orgeville et al., 2007; Ding et al., 2009; Greatbatch et al., 2012; Claus et al., 2016; Brandt et al., 2016). Resonant equatorial basin modes are low frequency standing

equatorial modes consisting of long equatorial Kelvin and Rossby waves (Cane and Moore, 1981). Depending on the gravity
wave speed and the basin geometry, each baroclinic mode has a characteristic resonance period. The semi-annual and annual
zonal flow variability in the equatorial Atlantic is attributed to the gravest basin mode for the second and the fourth baroclinic
mode, respectively (Brandt et al., 2016).

A realistic simulation of the narrow zonal current bands and their variability in the tropical Atlantic is still challenging.
While climate models are generally too coarse to fully resolve the tropical Atlantic current system, recent high-resolution ocean
general circulation models better represent the mean state of the zonal currents (Duteil et al., 2014). Still, distinct discrepancies
to ocean observation exists (Burmeister et al., 2020, 2019). Burmeister et al. (2020) showed a relationship between zonal
current and wind stress curl variability suggesting that it is important to resolve fine wind stress curl patterns to simulate the
narrow-banded zonal current system in the tropical Atlantic.

In this study we investigate how two different forcing products with different spatial and temporal resolution impact the
mean state and variability of the narrow-banded zonal current system in the tropical Atlantic. The forcing products are the
well-established but discontinued Coordinated Ocean-Ice Reference Experiments (CORE) v2 (Large and Yeager, 2009) and
its successor, the JRA55-do surface dataset (Tsujino et al., 2018). The simulations are performed with a global ocean model
covering the tropical Atlantic Ocean at eddying resolution, INALT20, which has the capability to resolve the complex zonal
current system in the tropical Atlantic. Furthermore, we have access to over ten years of velocity observations which period
is now covered by the JRA55-do forcing. This allows for a direct comparison between model and observations which is not
possible for simulations forced by CORE.

## 2   Data and Methods

In this section we describe the data and methods used in this paper. In summary, we compare two simulations with a high-
resolution global ocean circulation model forced by two different atmospheric products, the Coordinated Ocean-Ice Reference
Experiments (CORE) v2 dataset (Griffies et al., 2009) and the JRA55-do surface dataset v1.4.1 (Tsujino et al., 2018). We
calculate current transports for the eastward flowing EUC, NEUC, SEUC and NECC and the westward flowing nSEC utilizing
an algorithm which is following the current cores (Hsin and Qiu, 2012; Burmeister et al., 2019). The model results are compared
to shipboard hydrographic and velocity observation along 23°W (e.g. Brandt et al., 2015; Hahn et al., 2017; Burmeister et al.,
2020) and 35°W (Hormann and Brandt, 2007; Tuchen et al., 2022a) as well as current transport time series derived from
moored observations at 1.2°S to 1.2°N, 23°W (Brandt et al., 2021) and 5°N, 23°W (Burmeister et al., 2020). Furthermore, we
perform a modal decomposition of the simulated zonal velocity field and briefly introduce the equations used to calculate the
Sverdrup stream function, Ekman transport and an index for the activity of tropical instability waves (Lee et al., 2014; Olivier
et al., 2020; Perez et al., 2012; Tuchen et al., 2022b).

## 2.1 High-resolution Global Ocean Circulation Model INALT20

Our analyses are based on 5-daily averaged output of the global ocean circulation model INALT20. In INALT20 a $1/20°$ nest covering the South Atlantic and the western Indian oceans between 70°W and 70°E and the northern tip of the Antarctic Peninsula at 63°S to 10°N, is embedded into a global $1/4°$ host model (Schwarzkopf et al., 2019). The model is based on the Nucleus for European Modelling of the Ocean (NEMO) v3.6 code (the NEMO team, 2016) incorporating the Louvain-La-Neuve sea-ice model version 2 using a viscous–plastic rheology (LIM2-VP; Fichefet and Maqueda, 1997). A global configuration with tripolar grids, named ORCA025, is used as host to build the regionally finer resolved configuration realised by the AGRIF (Adaptive Grid Refinement in Fortran) library (Debreu et al., 2008). This set up allows for two-way interactions: the host not only provides boundary conditions for the nest but also receives information from the nest.

The model configuration has a vertical grid with 46 z levels varying in vertical grid size from 6m at the surface to 250 m in the deepest layers, resolving the first baroclinic mode (Stewart et al., 2017; Schubert et al., 2019) which is needed for the representation of the major baroclinic currents. The same vertical grid has proven to be an appropriate choice for simulations with model configurations up to $1/20°$ horizontal resolution (e.g. Böning et al., 2016; Behrens et al., 2017). The bottom topography is represented by partial steps (Barnier et al., 2006) with a minimum layer thickness of 25 m.

In this study we use two hindcast simulations which are forced with two different forcing products for the period 1958 to 2009 and 2019, respectively. The two hindcast simulations are preceded by a 30-year spin-up integration. The spin-up integration is initialised with temperature and salinity from the World Ocean Atlas (Levitus et al., 1998, with modifications in the polar regions from PHC; Steele et al., 2001) and an ocean at rest. The spin-up is forced by interannually varying atmospheric boundary conditions from 1980 to 2009 using CORE.

The well-established but discontinued CORE forcing covers the period 1948 to 2009 (Griffies et al., 2009) and builds on NCEP/NCAR reanalysis data merged with satellite-based radiation and precipitation, employing a set of parameter corrections to minimize global flux imbalances. Prior to the satellite era, CORE does not contain realistic time-varying radiation and precipitation fluxes as climatological values are used to fill in missing years (Large and Yeager, 2009). It has a horizontal resolution of 2°x2° and temporal resolution of 6-hours. CORE is limited by the relatively coarse spatial and temporal resolution and was discontinued in 2009 not covering the most recent decade of observations. Additionally, multi-decadal variability in this data set might be problematic as it includes NCEP winds known to exhibit spurious multidecadal wind variability (Fiorino, 2000; He et al., 2016; Hurrell and Trenberth, 1998). In the following we refer to the model simulation forced with the CORE forcing as $CORE_{sim}$.

The second forcing product is the more recent JRA55-do surface dataset v1.4.1 (Tsujino et al., 2018), which we refer to as JRA in the following. It is based on the 55-year Reanalysis project (JRA-55; e.g. Kobayashi et al., 2015) conducted by the Japan Meteorological Agency (JMA). This dataset stands out due to its higher horizontal ( 55 km) and temporal resolution (3 h) which now covers the entire observational period (1958 to 2018). Similar to CORE the surface fields from an atmospheric reanalysis are adjusted relative to reference datasets. The downwelling radiative fluxes and precipitation are based on reanalysis

products in contrast to CORE, which uses satellite observations. In the following we refer to the simulation forced with JRA as $\text{JRA}_{sim}$.

## 2.2 Shipboard observations

The meridional ship sections of velocity, hydrography and oxygen used in this study are an extension of the data set used by Burmeister et al. (2020). The data set consists of 31 velocity sections as well as 22 hydrographic and oxygen sections which were obtained during cruises along 21°W to 28°W between 2000 to 2018 (Table A1). Most sections are along 23°W between 6°S and 14°N and vertically extend from the surface to 600 m or 800 m.

Velocity data were acquired by vessel-mounted Acoustic Doppler Current Profilers (vm-ADCPs). Vm-ADCPs continuously record velocities throughout a ship section and the accuracy of 1-h averaged data is better than 2-4 cm s$^{-1}$ (Fischer et al., 2003). Hydrographic and oxygen data obtained during CTD casts were typically taken on a uniform latitude grid with half-degree resolution. The data accuracy for a single research cruise is generally assumed to be better than 0.002°C, 0.002 and 2 μmol kg$^{-1}$ for temperature, salinity, and dissolved oxygen, respectively (Hahn et al., 2017). The single velocity, hydrographic and oxygen ship section are mapped on a regular grid (0.05° latitude × 10 m) and are smoothed by a Gaussian filter (horizontal and vertical influence (cutoff) radii: 0.05° (0.1°) latitude and 10 m (20 m), respectively). The single sections are temporally averaged at each grid point to derive mean sections, which are again smoothed by the Gaussian filter.

To derive a second observational estimate for the mean current strength in the western basin, we additionally use 16 velocity and hydrographic ship sections along 35°W from 1990 to 2006 (Table A2). This dataset was used by Hormann and Brandt (2007) and Tuchen et al. (2022a). Note that shipboard velocity observations do not cover the upper most water layers. This is why all ship sections are limited to the shallowest common water depth, which is 30 m. This is also the upper limit used for any transport estimation of surface currents derived from shipboard observations.

## 2.3 Path following transport estimation

Transports of the zonal currents at a given longitude in the tropical Atlantic are estimated using the model output and shipboard observations following the approach of Hsin and Qiu (2012). First the central position $Y_{CM}$ of the current is estimated using the concept of centre of mass:

$$Y_{CM}(t) = \frac{\int_{Z_l}^{Z_u} \int_{Y_S}^{Y_N} y\, u(y,z,t)\, dy\, dz}{\int_{Z_l}^{Z_u} \int_{Y_S}^{Y_N} u(y,z,t)\, dy\, dz}, \tag{1}$$

where $y$ is latitude, $u$ is zonal velocity, $z$ is depth, $t$ is time, $Z_u$ ($Z_l$) is upper (lower) boundary of the flow defined as the depth of specific values of potential density (if not otherwise stated), and $Y_N$ ($Y_S$) is the northern (southern) limit of the current core.

Now the eastward velocity is integrated within a box whose meridional range is given by $Y_{CM}(t)$ and half mean width $W$ of the flow:

**Table 1.** Parameters for along pathway algorithm (Equ. 1 and 2)

| | EUC | NEUC | SEUC | NECC | nSECu | nSECl |
|---|---|---|---|---|---|---|
| $Z_u$ | $0\,\mathrm{kg\,m^{-3}}$ | $24.5\,\mathrm{kg\,m^{-3}}$ | $24.5\,\mathrm{kg\,m^{-3}}$ | $0\,\mathrm{kg\,m^{-3}}$ | $0\,\mathrm{kg\,m^{-3}}$ | $24.5\,\mathrm{kg\,m^{-3}}$ |
| $Z_{uobs}$ | $30\,\mathrm{m}$ | $24.5\,\mathrm{kg\,m^{-3}}$ | $24.5\,\mathrm{kg\,m^{-3}}$ | $30\,\mathrm{m}$ | $30\,\mathrm{m}$ | $24.5\,\mathrm{kg\,m^{-3}}$ |
| $Z_l$ | $26.8\,\mathrm{kg\,m^{-3}}$ | $26.8\,\mathrm{kg\,m^{-3}}$ | $27.0\,\mathrm{kg\,m^{-3}}$ | $24.5\,\mathrm{kg\,m^{-3}}$ | $24.5\,\mathrm{kg\,m^{-3}}$ | $26.8\,\mathrm{kg\,m^{-3}}$ |
| $Y_S$ | $1.2°\mathrm{S}$ | $3.5°\mathrm{N}$ | $6°\mathrm{S}$ | $4°\mathrm{N}$ | $0°$ | $0°$ |
| $Y_N$ | $1.2°\mathrm{N}$ | $6.0°\mathrm{N}$ | $4°\mathrm{S}$ | $10°\mathrm{N}$ | $5°\mathrm{N}$ | $4°\mathrm{N}$ |
| $W$ | $2°$ | $2°$ | $2°$ | $3.5°$ | $2°$ | $2°$ |
| $Y_{CM} - W$ | $3°\mathrm{S}$ | $2.5°\mathrm{N}$ | $6°\mathrm{S}$ (model $7°\mathrm{S}$) | $2.5°\mathrm{N}$ | $0°\mathrm{S}$ | $0°$ |
| $Y_{CM} + W$ | $2.5°\mathrm{N}$ | $8°\mathrm{N}$ | $3°\mathrm{S}$ | $10°\mathrm{N}$ | $6°\mathrm{N}$ | $5°\mathrm{N}$ |

$Z_u$ ($Z_l$) is upper (lower) boundary of the flow defined as the depth of specific values of potential density (if not otherwise stated), and $Y_N$ ($Y_S$) is the northern (southern) limit of the current core, $W$ is half mean width of current, $Y_{CM} + W$ ($Y_{CM} - W$) is the northern (southern) absolute limit for the flow integration. Note, as the moored and shipboard observations do not cover the upper water layer, we choose the upper boundary of the flow $Z_{Uobs}$ to be 30 m, the shallowest common depth of all observations.

$$INT(t) = \int_{Z_l}^{Z_u} \int_{Y_{CM}-W}^{Y_{CM}+W} u(y,z,t)\, dy\, dz \tag{2}$$

The parameters chosen for each current are listed in table 1.

## 2.4   Moored transport time series

We use long term observational transport time series estimated for the EUC by Brandt et al. (2021, 2014) and for the NEUC by Burmeister et al. (2020) to validate the model simulations. Transport time series of the EUC and NEUC are reconstructed from moored velocity observations at 0°N/23°W (May 2005-Sep 2019) and 5°N/23°W (Jul 2006-Feb 2008, Nov 2009-Jan 2018), respectively.

Horizontal velocity data were acquired using moored ADCPs. At the equator the upper water column was observed by one 300 or 150 kHz upward-looking ADCP between 100 to 230 m depth and another 75 kHz ADCP either downward-looking from just below the upper instrument or upward-looking from 600 to 650 m depth. Apart from a period between 2006 and 2008, when the upper instrument failed, the velocity measurements cover the whole depth range of the EUC. At 5°N either a downward- (Jul 2006-Feb 2008) or upward-looking (Nov 2009-Jan 2018) 75 kHz ADCP was installed. The upper measurement range of the 5°N ADCPs varies between 65 and 75 m which means that the upper 10 m of the NEUC is not always covered. This is accounted for in the model derived transport estimation when compared to observation. The effect of tides is removed from the moored velocity data by a 40-h low-pass Butterworth filter and subsampling to a regular 12-h time interval. The short-term variability of the tropical Atlantic exceeds the measurement accuracy of the different ADCPs and errors in the ADCP compass calibrations from different mooring periods are expected to be unsystematic (Brandt et al., 2021).

The EUC transport time series is estimated by regressing spatial variability patterns derived from shipboard observations onto the moored velocity time series at 0°N/23°W (Brandt et al., 2014, 2016, 2021). Eastward velocities ($u > 0$) of the reconstructed latitude-depth-sections (30-300 m depth and 1.2°S–1.2°N) are integrated to obtain the EUC transport. The root mean square differences for the EUC transport reconstruction using the equatorial mooring and the transport derived from the shipboard observations is 1.29 Sv (Brandt et al., 2014).

The NEUC transport time series is estimated from shipboard and moored velocity observations using the optimal width method (Burmeister et al., 2020). First, eastward velocities ($u > 0$) of shipboard observations are latitudinally integrated between 65 and 270 m depth, 4.25°N and 5.25°N. To reconstruct the latitudinally integrated velocities ($U(z)$), an optimal latitude range needs to be found by regressing $U(z)$ onto the shipboard eastward velocity profile at the mooring position. The moored velocity profiles are multiplied by the optimal latitude range (0.88°) and finally depth-integrated to obtain the NEUC transport time series. The root mean square difference of the reconstructed NEUC transport from the shipboard observations is 0.52 Sv (Burmeister et al., 2020).

Note that the reconstructed transport represents the current transport integrated over a fixed box. To compare transport from model output and moored observations at 23°W, we calculate the transport for the EUC and NEUC from model output as integral of eastward velocity in the respective box (EUC: 30 m-300 m, 1.2°S-1.2°N; NEUC: 60 m-270 m, 4.25°N-5.25°N).

## 2.5 Modal Decomposition

We decompose the velocity field of the two model simulations using vertical structure functions $\hat{p}_n(z)$ obtained from a mean buoyancy frequency profile derived from observations (Brandt et al., 2016). Following the approach of Claus et al. (2016), we derive $\hat{p}_n(z)$ from a mean buoyancy frequency profile obtained from 70 shipboard CTD profiles (Table A1). To obtain the mean buoyancy frequency profile we use CTD profiles with a minimum depth of 1200 m within a 1°-wide squared box centred at 23°W, 0°N. We bin-average the individual temperature and salinity profiles to a uniform 10-m vertical grid with a maximum depth of 4500 m and calculate a buoyancy frequency profile for each cast separately which are then average to obtain the mean buoyancy frequency profile. It is important to note that baroclinic modes are only orthogonal if the velocity data is covering the complete upper 4500 m depth. Missing data as typical for shipboard or moored data reduce the orthogonality and introduce uncertainties in the calculation. However, consistent results between studies provide some confidence in the chosen approach (e.g. Brandt et al., 2016; Claus et al., 2016; Kopte et al., 2018).

The gravity wave speed of the first five baroclinic modes derived from observations is shown in table 2. We also derive the vertical structure functions from mean buoyancy frequency profiles using model output from $JRA_{sim}$ and $CORE_{sim}$. For the gravity wave speed, the two simulations are in good agreement both with each other and with observations.

To estimate the contribution of the first five modes to the annual and semi-annual cycle of the zonal velocity field in the tropical Atlantic (10°S-10°N) we use the orthogonality between functions. We fit the vertical normal baroclinic modes and temporal harmonics with reduction operations as follows.

**Table 2.** Gravity wave speed of the first 5 baroclinic modes of the gravest basin mode using squared buoyancy frequencies within a $1°$-wide squared box centred at 23°W, 0°N using CTD profiles as well as model output from $JRA_{sim}$ and $CORE_{sim}$.

|  | mode 1 | mode 2 | mode 3 | mode 4 | mode 5 |
|---|---|---|---|---|---|
| CTD | 2.51 | 1.40 | 0.98 | 0.76 | 0.57 |
| $JRA_{sim}$ | 2.53 | 1.43 | 1.05 | 0.81 | 0.58 |
| $CORE_{sim}$ | 2.51 | 1.42 | 1.04 | 0.80 | 0.57 |

Let $\hat{p}_n(z)$ be vertical normal (baroclinic) modes with

$$\int \mathrm{d}z\, \hat{p}_n(z) \cdot \hat{p}_m(z) = \delta_{n,m} \tag{3}$$

and $h_{T,\tau}(t)$ be temporal harmonic modes with period $T$ and phase $\tau$ which fulfil

$$\int \mathrm{d}t\, h_{T,0}(t) \cdot h_{T/2,0}(t) = 0 \tag{4}$$

$$\int \mathrm{d}t\, h_{T,0}(t) \cdot h_{T,0}(t) = 1 \tag{5}$$

Then, we could compose a signal $s(t,z)$ with different normal modes each having a separate annual and semi-annual cycle as:

$$s(t,z) = \sum_n \alpha_n^a \cdot \hat{p}_n(z) \cdot h_{365d,\tau_n^a}(t) + \sum_n \alpha_n^s \cdot \hat{p}_n(z) \cdot h_{365d/2,\tau_n^s}(t) \tag{6}$$

where $\alpha_n^a$ is the amplitude of the annual cycle of the baroclinic mode $n$, $\alpha_n^s$ is the amplitude of the semi-annual cycle of the

245 baroclinic mode $n$, $\tau_n^a$ is the phase shift of the annual cycle of baroclinic mode $n$, and $\tau_n^s$ is the phase shift of the semi-annual cycle of baroclinic mode $n$.

The time-variability of the baroclinic mode $n$ can be diagnosed using a depth integral:

$$\alpha_n^a \cdot h_{365d,\tau_n^a}(t) + \alpha_n^a \cdot h_{365d/2,\tau_n^a}(t) = \int \mathrm{d}z\, b_n(z) \cdot s(t,z) \equiv s_n(t) \tag{7}$$

The phase and amplitude of $s_n(t)$ can be diagnosed by projecting a time series covering and integer number of years on a

250 normalised annual $\mathrm{e}^{i2\pi/365d \cdot t}$ or semi-annual $\mathrm{e}^{i4\pi/365d \cdot t}$:

$$\alpha_n^a \propto \left| \int \mathrm{d}t\, \mathrm{e}^{i2\pi/365d \cdot t} s_n(t) \right|, \qquad\qquad \alpha_n^s \propto \left| \int \mathrm{d}t\, \mathrm{e}^{i4\pi/365d \cdot t} s_n(t) \right| \tag{8}$$

$$\tau_n^a = \arg\left( \int \mathrm{d}t\, \mathrm{e}^{i2\pi/365d \cdot t} s_n(t) \right), \qquad\qquad \tau_n^s = \arg\left( \int \mathrm{d}t\, \mathrm{e}^{i4\pi/365d \cdot t} s_n(t) \right) \tag{9}$$

## 2.6 Sverdrup Balance

The Sverdrup balance relates the meridional volume transport in the ocean interior to the wind stress curl. It can be derived from the momentum balance between pressure gradient, Coriolis force and wind stress (Sverdrup, 1947). We calculate the Sverdrup stream function as follows:

$$\Psi = -\frac{1}{\rho_0 \beta} \left( \int_{x}^{x_0} \left( \hat{k} \nabla \times \tau \right) dx \right), \qquad \beta = \frac{\partial f}{\partial y} = \frac{2\Omega \cos(\phi)}{R_{Earth}} \tag{10}$$

where $\rho_0 = 1025 \, \text{kg} \, \text{m}^{-3}$ is the mean water density, $x_0$ refers to the west coast of Africa, $x$ is longitude, $(\hat{k} \nabla \times \tau)$ is the wind stress curl, $\Omega = 7.271 \cdot 10^{-5} \, \text{s}^{-1}$ is the angular velocity of the Earth rotation, $R_{Earth} = 6.37 \cdot 10^6 \, \text{m}$ is the radius of the Earth, $\phi$ is latitude. To estimate the contribution of Sverdrup dynamics to the zonal current transports we calculate the difference in the Sverdrup stream function $\Psi$ (Equ. 10) between the bounding latitudes of each current:

$$U_{\Psi} = \Psi_N - \Psi_S \tag{11}$$

Please note, the Sverdrup stream function represents the depth-integrated wind-driven flow field. For example, between 4°N and 6°N, the resulting zonal flow calculated from the Sverdrup stream function is distributed across several currents, the NECC at the surface and the NEUC below.

## 2.7 Ekman transport and subtropical cells

The wind driven STCs connect subtropical subduction regions with the tropical upwelling region (e.g., Schott et al., 2004; Tuchen et al., 2019) and can impact the strength of zonal currents in the tropical Atlantic (Rabe et al., 2008). The strength of the STCs is related to the Ekman divergence which is commonly defined as the difference of Ekman transport between 10°S and 10°N (Rabe et al., 2008; Tuchen et al., 2019). Assuming the upper branch of the STCs is governed by poleward Ekman transport we calculate it as follows:

$$T_E(x, y, t) = -\frac{1}{\rho_0} \frac{\tau_x(x, y, t)}{f(y)} \Delta x \tag{12}$$

where $\tau_x$ represents the zonal wind stress component and $\Delta x$ is the zonal grid spacing in the model simulation.

## 2.8 Tropical instability wave activity

Part of the NEUC and SEUC are thought to be driven by mesoscale eddies or vortices, among other tropical instability waves (TIWs, e.g. Jochum and Malanotte-Rizzoli, 2004; Assene et al., 2020). To see if a different intensity of TIWs exist between the two model simulations we calculate the TIW activity from the simulated meridional velocity field at 160 m depth. We first apply a 20-50 day bandpass filter, followed by a 4°-20° bandpass filter to the 5 daily meridional velocity field ($v'$) by using a

second order, zero-phase Butterworth bandpass filter. Then, we calculate the monthly standard deviation from the filtered data $(\sigma(v'))$. This is a well-established method for the analysis of TIWs (Lee et al., 2014; Olivier et al., 2020; Perez et al., 2012; Tuchen et al., 2022b). Finally we box average the monthly standard deviation of meridional velocity between 3°N and 7°N, 30°W and 10°W for the NEUC and 3°S and 7°S, 30°W and 0° for the SEUC (Fig. 7).

## 3    Results

To compare the two model simulations $\mathrm{CORE}_{sim}$ and $\mathrm{JRA}_{sim}$ we focus on quantities of or derived from the wind forcing as well as quantities of or derived from the simulated velocity field. In particular we compare zonal wind stress, wind stress curl, zonal velocity and zonal current transport and discuss it in terms of, among others, the Sverdrup stream function and meridional Ekman divergence derived from the wind forcings, and the resonant equatorial basin modes fitted to the simulated velocity field. We then compare the mean fields, seasonal variability and longer-term variability and trends. Where possible, we compare the simulations with observations.

### 3.1    Mean fields

$\mathrm{CORE}_{sim}$ and $\mathrm{JRA}_{sim}$ both represent the common large scale wind stress pattern in the tropical Atlantic (Fig. 1): The southeasterly trades cross the equator, leading to negative wind stress curl south of about 2°N spanning the full width of the Atlantic as well as in the eastern basin west of 20°W and south of 6°N. Positive wind stress curl occurs north of these regions. Between 2° and 5°N west of 20°W the wind stress curl is up to three times stronger in $\mathrm{CORE}_{sim}$ than in $\mathrm{JRA}_{sim}$ for the period 1980 to 2009. $\mathrm{JRA}_{sim}$ resolves much finer wind stress curl structures than $\mathrm{CORE}_{sim}$, especially along the western and eastern boundaries. Another important feature of the wind stress curl is the minimum at about 6°N which drives the sea level slope important for the NECC. This is much more pronounced in $\mathrm{JRA}_{sim}$ compared to $\mathrm{CORE}_{sim}$.

As a first measure to evaluate how this might impact the wind-driven current field in the tropical Atlantic, we calculate the Sverdrup stream function of the temporal averaged wind stress curl (contour lines in Fig. 1). The tropical gyre north and the equatorial gyre south of the NECC are clearly visible in the two simulations. In $\mathrm{CORE}_{sim}$ the tropical gyre extends further to the south, especially in the western basin, compared to $\mathrm{JRA}_{sim}$. In $\mathrm{JRA}_{sim}$ the mean position of the NECC near 6°N lines up with the zero crossing of the Sverdrup stream function between the two gyres. In $\mathrm{CORE}_{sim}$ this is the case east of 20°W, while the NECC is displaced northward of the zero-crossing west of it. In general, largest differences of the Sverdrup stream function between $\mathrm{JRA}_{sim}$ and $\mathrm{CORE}_{sim}$ occur north of the equator (Fig. 1c).

Next, we compare the mean zonal velocity field derived from repeated ship sections along 23°W and 35°W with the simulated mean zonal velocities along these latitudes in the two simulations (Fig. 2a-f). The off-equatorial zonal currents are known to be mostly in geostrophic balance (e.g Jochum et al., 2004; Brandt et al., 2010; Goes et al., 2013). This relationship is represented well in the mean zonal velocity sections with stronger currents associated with steeper sloping of isopycnals and vice versa. Interestingly, largest differences on the 23°W section between the simulations occur north of the equator within the region of the NECC, the NEUC and the nSEC. $\mathrm{CORE}_{sim}$ tends to overestimate the strength and vertical extent of these zonal

currents compared to $JRA_{sim}$ and observations. At 35°W, these currents are of similar strength in the two simulations and compare reasonably well to observations. The zonal variation in the differences between the two simulations is also visible in the current transport calculated using Equation 2 and the parameter from Table 1 (Fig. 2g-l). Please note, that due to the large vertical extent of the nSEC we calculate the transport for the upper part (nSECu, transports above the $24.5 \, \mathrm{kg \, m^{-3}}$ isopycnal) and the lower part (nSECl, transports below the $24.5 \, \mathrm{kg \, m^{-3}}$ isopycnal) of the nSEC separately. The transport of currents north of the equator from the two simulations diverge east of 35°W (NECC) or 30°W (NEUC, nSECl) with $CORE_{sim}$ producing stronger currents at 23°W (Fig. 2h,j,l). At 35°W, the two model simulations agree well with the observations for the NEUC and nSECl and $JRA_{sim}$ only for the EUC. At 23°W, the two simulations tend to overestimate the current transport compared to observations apart for the SEUC which observed transport is about twice as high as the simulated transports. In general, $CORE_{sim}$ simulates higher transports than $JRA_{sim}$.

To assess how much of the inter-simulation differences in the flow field can be attributed to the wind stress fields and the resulting Sverdrup transport, we use the depth integrated vorticity equation. Under Sverdrup balance and to leading order, it can be expressed as the balance of the linear advection term $\beta \rho_0 \int_{-H}^{0} v dz$ and the wind stress curl, where v is the simulated meridional velocity and $H = 500 \, \mathrm{m}$ is the depth of the active ocean layer of interest (Small et al., 2015). Please note, the balance requires an integration depth where the vertical velocity is zero. Given that the isopycnals along $500 \, \mathrm{m}$ are quite flat in the mean sections at 35°W and 23°W we assume that this criterion is approximately fulfilled for long-term means. Differences between the linear advection term and the wind stress curl show where the Sverdrup balance does not hold, for example at the western boundary (Fig. A1). When subtracting the wind stress curl from the linear advection term, the magnitudes in $JRA_{sim}$ and $CORE_{sim}$ compare better in the central basin while differences remain in the spatial pattern. The inter-simulation differences in the wind stress curl and the associated Sverdrup balance hence can explain only part of the difference found in the flow field north of the equator.

The off-equatorial subsurface currents (NEUC, SEUC) are suggested to be partly driven by the pull of upwelling within domes or at the eastern boundary (Furue et al., 2007, 2009; McCreary et al., 2002) and previous observational and model studies found a link between the upwelling regions in the Atlantic and the NEUC (Stramma et al., 2005; Hüttl-Kabus and Böning, 2008; Goes et al., 2013) as well as the SEUC (Doi et al., 2007). We box-averaged the temporal mean (1980-2009) of wind stress curl and Ekman pumping within the Guinea upwelling region (5°-10°N, 30°-15°W) and found them to be 1.5 time higher in $CORE_{sim}$ ($1.9 \cdot 10^{-8} \, \mathrm{N \, m^3}$, $0.8 \, \mu \mathrm{m \, s^{-1}}$) compared to $JRA_{sim}$ ($1.2 \cdot 10^{-8} \, \mathrm{N \, m^3}$, $0.5 \, \mu \mathrm{m \, s^{-1}}$). The zonally averaged, temporal mean (1980-2009) of the NEUC transport west of 25°W is also 1.5 time higher in $CORE_{sim}$ ($6 \, \mathrm{Sv}$) compared to $JRA_{sim}$ ($4 \, \mathrm{Sv}$). In contrast, the SEUC has a similar mean strength in the two simulations as do the box-averaged, temporal mean (1980-2009) of wind stress curl ($3.7 \cdot 10^{-8} \, \mathrm{N \, m^3}$) and Ekman pumping ($2.5 \, \mu \mathrm{m \, s^{-1}}$) in a subregion of the Angola Dome region (7.5°–4.5°S, 0.5°W–2.5°E) that has been linked to the SEUC by Doi et al. (2007). The comparison of current strength, wind stress curl and Ekman pumping in the upwelling domes between the two simulations suggests that the inter-simulation differences of the NEUC are likely due to differences in the wind stress curl and associated upwelling in the Gulf of Guinea. The good inter-simulation agreement of the SEUC transport fits well to the good agreement of the wind-driven upwelling in the Angola Dome found between the two simulations.

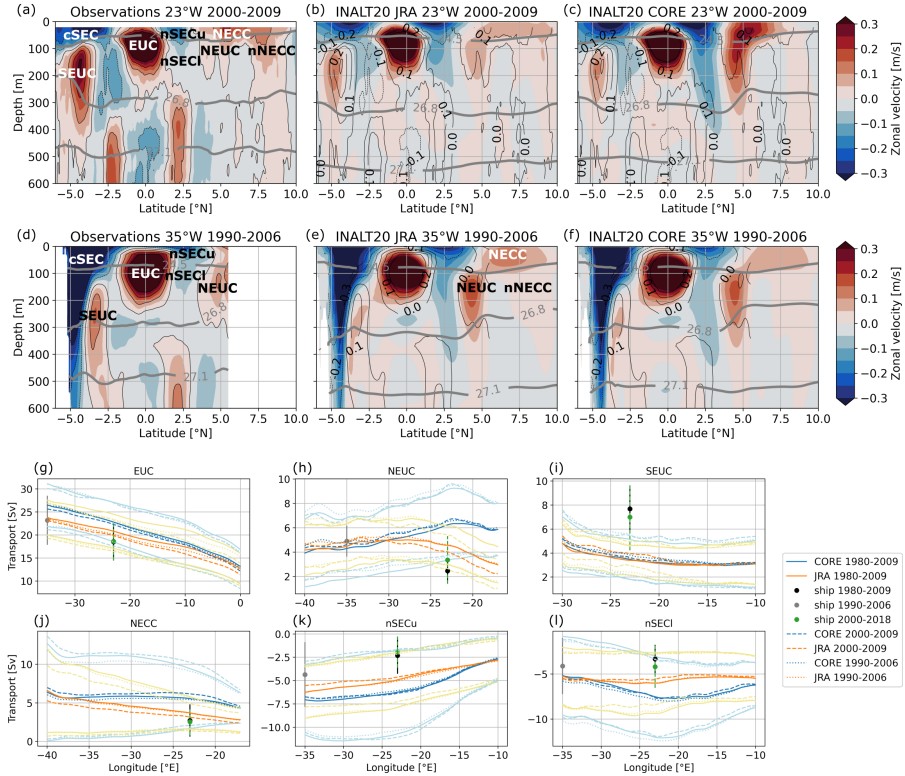

**Figure 2.** Mean zonal velocity along (a-c) 23°W (2000-2009) and (d-f) 35°W (1990-2006) from (a,d) observations, (b,e) JRA$_{sim}$ and (c,f) CORE$_{sim}$. Eastward velocities are positive (red), and westward are negative (blue). Grey thick contours mark potential density surfaces (kg/m$^3$), thin black contours in (a-f) mark observed velocities. (g-l) Temporal mean transport calculated for different periods (solid: 1980-2009; dashed: 2000-2009; dotted: 1990-2006) from 5-daily model output of CORE$_{sim}$ (blue lines) and JRA$_{sim}$ (orange lines) as well as ship sections (black dots 2000-2009, green dots 2000-2018, grey dots 1990-2006) using Equation 2 and the parameters listed in Table 1. The pastel blue and orange lines as well as the black, green and grey bars represent one standard deviation of model output and ship section in their respective temporal resolution.

The eastward flowing NECC has been also shown to be partly connected to the Guinea Dome (Stramma et al., 2005; Hormann et al., 2012; Stramma et al., 2016). Similar to the NEUC we find that the NECC is on average 1.5 times stronger in CORE$_{sim}$ (5.4 Sv) than in JRA$_{sim}$ (3.7 Sv) east of 30°W. Furthermore, the negative wind stress curl anomaly east of 23°W between 3°N and 5°N drives eastward Sverdrup flow at 5°N strengthening the NECC and NEUC in CORE (Fig. 1c). The 350 zonal transport resulting from the meridional Sverdrup transport between 4°N and 6°N ($U_\Psi$, Equ. 11) shows eastward flow in CORE$_{sim}$ which is 1 Sv (30°W) to 2.7 Sv (20°W) higher than in JRA$_{sim}$. Regarding the entire meridional extent of the NECC (3°-10°N), the mean current transport (JRA: 5.2 Sv, CORE: 5.7 Sv) and $U_\Psi$ (JRA: 5.5 Sv, CORE: 5.4 Sv) agree well at 35°W in the two simulations, while they start to diverge further east, with eastward $U_\Psi$ flow in CORE being up to 0.9 Sv (23°W) stronger then in JRA. The anomalous Sverdrup stream function also suggests that CORE drives a strong recirculation between

the nSEC and NECC/NEUC which agrees with the findings of Burmeister et al. (2019) and shows enhanced westward flow along the core position of the nSEC in CORE (Fig. 1c). However, the comparison between the wind stress curl and the linear advection term (Fig.A1) highlights that inter-simulation differences in the wind stress curl and associated Sverdrup transport can only partly explain inter-simulation differences in the flow field in that region. The surface flowing nSEC is mainly driven by the equatorial easterlies. The mean zonal wind stress (1980-2009) box-averaged above the SEC region (0°-5°N, 35°-15°W) is 1.2 times stronger in $CORE_{sim}$ compared to $JRA_{sim}$, as are the zonally averaged current transports for both, the nSECu and nSECl.

One of the reasons for the inter-simulation discrepancies might be the coarser spatial resolution of the CORE forcing. Due to its high spatial resolution, the JRA forcing is thought to better resolve fine wind stress curl structures. To get an idea how much the spatial resolution matters, we bin-average the wind stress fields of the two simulations to a spatial resolution of $2°×2°$ (Fig. A5). Compared to $JRA_{sim}$, $CORE_{sim}$ still shows increased positive wind stress curl along the western boundary, in the central basin along 5°N, within the Guinea Dome region and along Northwest Africa. However, the difference of the Sverdrup stream function between the coarse resolution fields of $JRA_{sim}$ and $CORE_{sim}$ (Fig. A5f) do not show any small scale features visible in the high-resolution fields above the nSEC and NECC/NEUC region with differences in the Sverdrup stream function of 0.5 Sv to 1.5 Sv east of 30°W (Fig.1c).

The EUC is mainly driven by the easterly winds along the equator (Pedlosky, 1987; Wacongne, 1989). However, Arhan et al. (2006) showed that in the absence of the equatorial zonal wind during winter and spring, EUC transport can be remotely forced by the wind stress curl between 2°S and 2°N connecting it to the western boundary currents. The mean zonal wind stress (1980-2009) in $CORE_{sim}$ along the equator is stronger (-0.034 N m$^{-2}$ with a standard deviation of $\pm0.012$ N m$^{-2}$) than in $JRA_{sim}$ (-0.027 N m$^{-2}$ with a standard deviation of $\pm0.011$ N m$^{-2}$). The inter-model difference in the mean wind stress can be one reason why the EUC transports are stronger in CORE compared to JRA. Another process impacting the strength of the EUC is the strength of the STCs. The different strength of the trade winds between the forcings may lead to a different strength in the poleward Ekman transport forming the upper branch of the STCs which again can cause different strength in the EUC (Rabe et al., 2008). The strength of the STCs is related to the meridional Ekman divergence which is quantified as the divergence of the Ekman transport (Equ. 12) between 10°S ($JRA_{sim}$ -9.4 Sv, $CORE_{sim}$ -11 Sv) and 10°N ($JRA_{sim}$ 9.3 Sv, $CORE_{sim}$ 11.4 Sv). The calculated meridional Ekman divergence for the two simulations ($JRA_{sim}$ 18.7 Sv, $CORE_{sim}$ 22.4 Sv) is within the range of estimates derived for different wind products in Tuchen et al. (2019, 20.4$\pm$3.1 Sv). We find that the meridional Ekman divergence in $CORE_{sim}$ is 3.7 Sv larger than in $JRA_{sim}$, which can contribute to a stronger mean EUC transport in $CORE_{sim}$ compared to $JRA_{sim}$. Furthermore, at 35°W (23°W), the difference in the mean eastward Sverdrup transport between 2°S and 2°N for the period 1980 to 2009 is 2.2 Sv (0.2 Sv) higher in $CORE_{sim}$ than in $JRA_{sim}$, which might also contribute to a stronger EUC in $CORE_{sim}$, especially west of 20°W.

## 3.2 Seasonal cycle

The seasonal cycle in the tropical Atlantic circulation is dominated by the meridional migration of the ITCZ and concomitant changes in the wind field (e.g., Xie and Carton, 2004). In the following we investigate how differences in the seasonal cycle of

the wind forcings impact the seasonal cycle of the zonal currents. First, we show the main patterns of the seasonal cycle of the
390 wind forcing by fitting the annual harmonic to the zonal wind stress and wind stress curl for the period 1980 to 2009 (Fig. 3).
Then, we describe the seasonal cycle of simulated path-following current transports (Equ. 2) for the same period (Fig. 4). This
is followed by a model validation, where we focus on the transports of the EUC and NEUC for the period 2000-2018 when
we have moored transport observations available which are calculated within a fixed box for consistency between model and
observations (Fig. 5). Finally, we investigate the link of the seasonal cycle between the wind forcing and the velocity field in
the two simulations under the aspect of resonant equatorial basin modes (Fig. 6).

The large-scale pattern of zonal wind stress and wind stress curl amplitudes of the annual harmonic cycle are similar in
the two simulations while the $\text{CORE}_{sim}$ produces much higher amplitudes compared to $\text{JRA}_{sim}$ (Fig. 3). Again, the wind
stress curl is characterised by fine spatial structures in $\text{JRA}_{sim}$ which are not present in $\text{CORE}_{sim}$. Largest differences in zonal
wind stress and wind stress curl occur north of the equator in the eastern basin. The spatial pattern of the phase of the annual
harmonic differs between the simulations for zonal wind stress. This leads to phase shifts between the simulations of 0 to 6
months depending on longitude and latitude. The spatial pattern of the phase of the wind stress curl agrees better between the
two simulations. However, between 4°S and 4°N, west of 20°W the phase is very homogeneous in $\text{JRA}_{sim}$, while we see a
change of phase with longitude of up to 6 months in $\text{CORE}_{sim}$. The annual harmonic amplitude of zonal wind stress along
the equator is much larger in $\text{CORE}_{sim}$ compared to $\text{JRA}_{sim}$. Before we investigate how these differences in the wind forcing
impact the zonal current variability, we first describe and validate the seasonal cycle of simulated zonal current transports.

Compared to $\text{JRA}_{sim}$, $\text{CORE}_{sim}$ exhibits a stronger annual and semi-annual cycle of the zonal current transports, especially
at 23°W, except for the SEUC (solid lines in Fig. 4). Aligning with the results for the mean current strength (Fig. 2), the
seasonal variability of the SEUC is very similar in the two simulations and the model tends to underestimate the SEUC
strength compared to observations (Fig. 4g). At 23°W, for the EUC and nSECu, the amplitude of the annual cycle in $\text{JRA}_{sim}$
peaks in late boreal spring/early summer, two to three months earlier than in $\text{CORE}_{sim}$. For the other currents and for the phase
of the semi-annual cycle, the two simulations are in good agreement at 23°W. In general, we find a better agreement between
the two simulations regarding the phase and amplitude of the seasonal current transports at 35°W (dashed lines in Fig. 4) than
at 23°W.

To get a better understanding of how realistic the model simulates the seasonal cycle of the currents, we compare the seasonal
cycle of the simulated currents with moored transport time series available for EUC and NEUC at 23°W only. Note that the
reconstructed transports from moored observations are integrated in a fixed box. To compare transports derived from model
output and moored observations at 23°W, we calculate transports for the EUC and the NEUC from model output as integral of
eastward velocity in the respective box (EUC: 30 m-300 m, 1.2°S-1.2°N; NEUC: 60 m-270 m, 4.25°N-5.25°N). Furthermore,
the transports for the seasonal, annual and semi-annual cycles in the following are calculated for a shorter time period covering
the time period of observations if possible (see caption of Fig. 5 for more detail).

At 23°W, the $\text{CORE}_{sim}$ better represents the phase of the annual cycle of the EUC (Fig. 5e-g). We find a three-month phase
shift of the annual cycle in $\text{JRA}_{sim}$ compared to observations. The phase shift of the annual cycle between the $\text{CORE}_{sim}$ and
the observations is one month (Fig. 5f). For the semi-annual harmonic $\text{JRA}_{sim}$ seems to better represent the amplitude of the

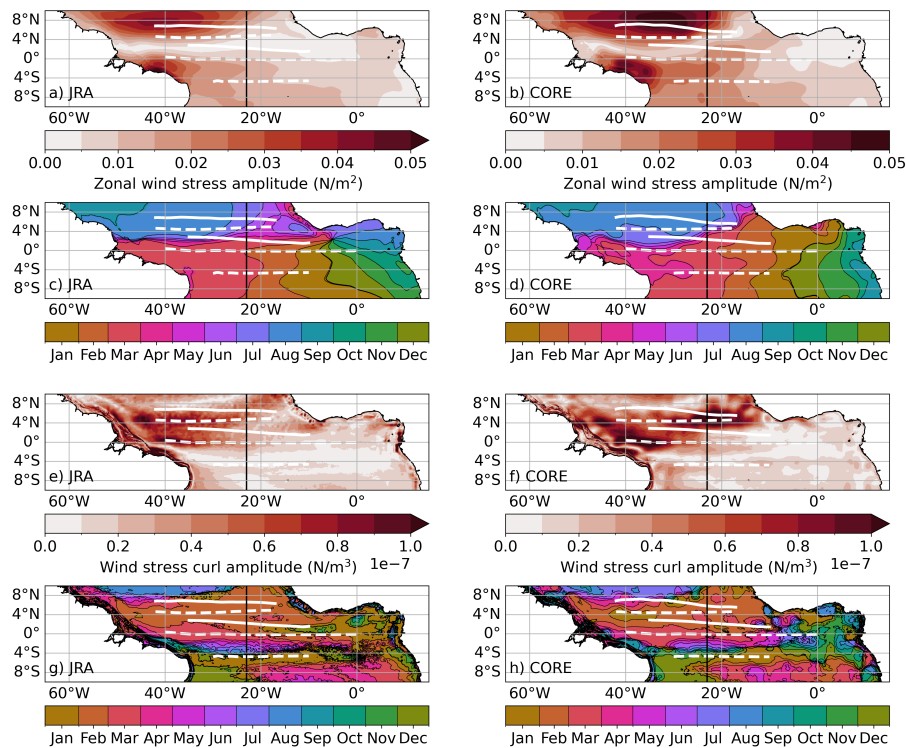

**Figure 3.** Amplitude (a,b,e,f) and phase (c,d,g,h) of annual harmonic fitted to monthly mean 1980-2009 climatology of zonal wind stress (a-d) and wind stress curl (e-h) from JRA$_{sim}$ (left) and CORE$_{sim}$ (right) for 1980-2009. Zonal white lines mark the mean latitude ($Y_{CM}$, Equ. 1) of the simulated surface (solid) and subsurface (dashed) currents for the respective periods. The black vertical lines mark 23°W. Please note the phase is given as the month of the year when the corresponding amplitude is maximum.

observations, while both simulations show a phase shift of about one month compared to observations (Fig. 5g). Within the
chosen parameters (Tab. 1), JRA$_{sim}$ cannot reproduce the EUC intensification in boreal fall which seems to be related to the annual cycle peaking in boreal summer. Note that increasing the half mean width $W$ in equation 2 from 2° to 3° results in a 2 Sv increase of the seasonal cycle of EUC transport in boreal fall (2006-2018) and the fitted annual harmonic is maximum at the end of July (dashed lines in Fig. 5e-g).

     The representation of the NEUC transport variability is more realistic in JRA$_{sim}$ compared to CORE$_{sim}$ (Fig. 5). JRA$_{sim}$
better captures the sporadic intraseasonal fluctuations of the NEUC which is dominating the NEUC variability in the observations (Burmeister et al., 2020). In CORE$_{sim}$ the NEUC variability is dominated by a strong seasonal cycle instead (amplitude of 1.8 Sv) even though the spectral analysis of the NEUC in CORE$_{sim}$ is more energetic on intraseasonal timescale compared to JRA$_{sim}$ and observations. Compared to observations, the seasonal cycle of the NEUC in JRA$_{sim}$ is more realistic but still too strong (JRA$_{sim}$ 0.6 Sv vs observations 0.2 Sv). JRA$_{sim}$ produces a NEUC flow maximum in April to May which
is not visible in the observations, but both, the simulated and observed seasonal NEUC cycle show a minimum in boreal fall.

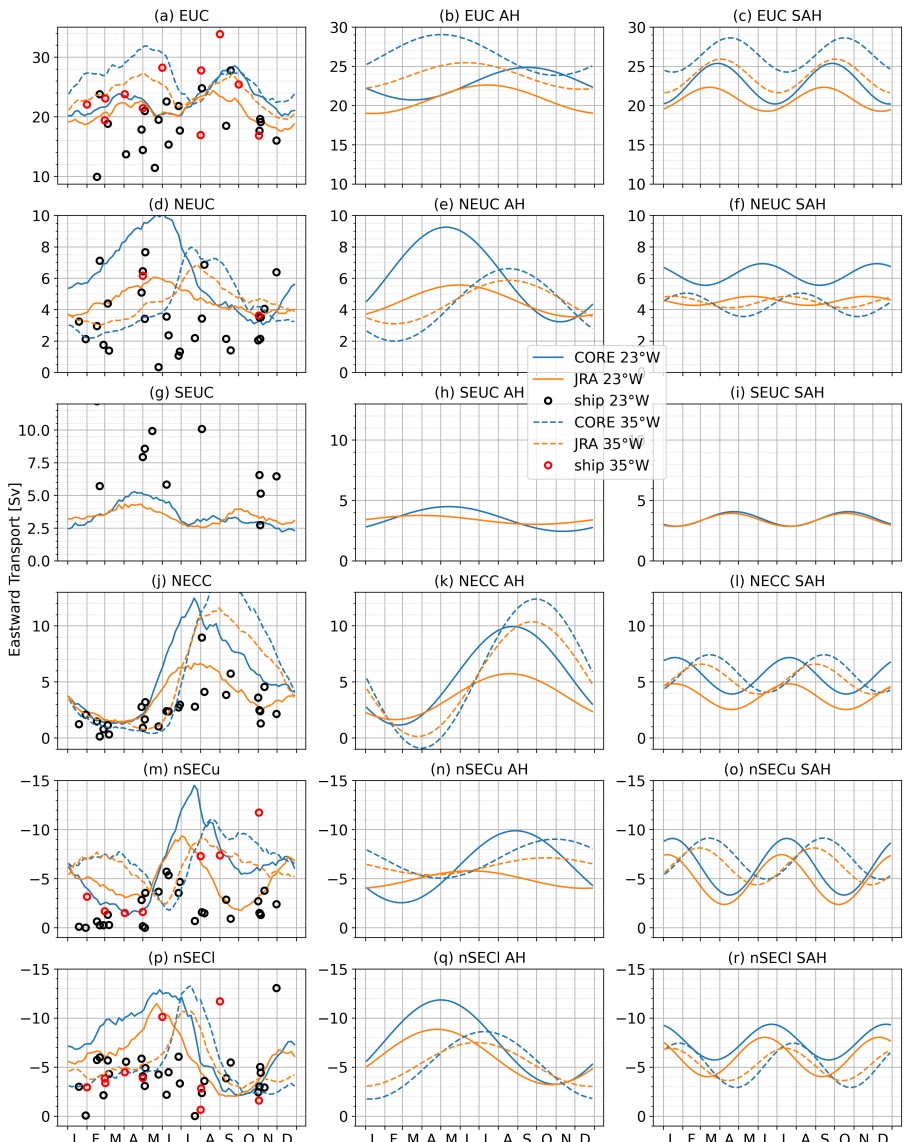

**Figure 4.** (left) Seasonal cycle, (middle) annual harmonic and (right) semi-annual harmonic fitted to the transport time series (Equ. 2) estimated from $JRA_{sim}$ (orange lines), $CORE_{sim}$ (blue lines) at $23°W$ (solid lines) and $35°W$ (dashed lines) averaged over the period 1980 to 2009. Black and red circles in the left column mark transports estimated from ship sections along $23°W$ (black) and $35°W$ (red).

Burmeister et al. (2020) suggested that the NEUC fluctuations might be triggered by Rossby waves which can alter the weak eastward flow of the NEUC. They showed among others that small scale wind stress curl anomalies off the coast of Liberia lead NEUC fluctuations by 1-2 months. Our results suggest that the JRA wind forcing seems to better resolve mechanisms dominating NEUC variability, while $CORE_{sim}$ seems not to be able to resolve them (Fig. 3e and f). Instead, the annual cycle

of the winds stress curl in $CORE_{sim}$ shows high amplitudes between 4°N and 8°N west of 30°W which might contribute to the strong annual cycle of the NEUC. In $CORE_{sim}$ the amplitude of the annual cycle of the wind stress curl averaged in that region (4°-8°N, 30°-15°W) is twice as strong as in $JRA_{sim}$.

The semi-annual and annual zonal flow variability along the equator is attributed to the resonance period of the gravest basin mode for the second and the fourth baroclinic mode, respectively (Brandt et al., 2016). We perform a baroclinic modal decomposition of the zonal velocity field in the two model simulations to investigate possible resonances and the dynamical response of the ocean to the two wind forcings. We fit the annual and semi-annual harmonic to the first five baroclinic modes. In both simulations, we find high amplitudes of the annual harmonic along the equator for baroclinic mode four (Fig. 6) and three (Fig. A3). Along the equator, velocity amplitudes for the annual cycle of baroclinic mode three and four in $CORE_{sim}$ are up to $2.5\,\mathrm{cm\,s^{-1}}$ higher than in $JRA_{sim}$ with largest difference occurring between 30°W and 20°W for baroclinic mode three (Fig. A3). This agrees with Brandt et al. (2016) who found that the third mode in their model simulation also forced with CORE was enhanced compared to observations. Along the equator the phase of the maximum velocity of the annual cycle differs between the two simulations (Fig. 6). Between 0° and 40°W along the equator (±0.5°), maximum velocities in the $JRA_{sim}$ occur on average 22 days earlier than in the $CORE_{sim}$ with a standard deviation of 6 days. For the semi-annual cycle the differences are less distinct (Fig. A2) in agreement with the EUC transport time series (Fig. 5). As the phase velocities of the first five baroclinic modes in the two simulations are similar (Tab 2) it is likely that the differences are mainly due to the annual cycle of the wind forcing.

Along 5°N within the NEUC region, the amplitude of the annual cycle of the fourth baroclinic mode is slightly higher in the $CORE_{sim}$ and extends further east compared to the $JRA_{sim}$ (Fig. 6). Largest differences in annual cycle amplitudes exist for the first two baroclinic modes just north of the NEUC mean position and south of the nSEC mean position (Fig. A3). This might be one factor why we find a strong annual cycle for the NEUC in $CORE_{sim}$ and a weak annual cycle in $JRA_{sim}$.

Parts of the NEUC and SEUC are thought to be driven by mesoscale eddies or vortices (e.g. Jochum and Malanotte-Rizzoli, 2004; Assene et al., 2020). Among others the Eliassen-Palm flux of tropical instability waves (TIWs) is thought to maintain the eastward subsurface currents against dissipation (Jochum and Malanotte-Rizzoli, 2004). Assene et al. (2020) described how westward propagating mesoscale vortices (e.g. TIWs) east of 20°W can create high potential vorticity gradients in the mean fields which are associated with the NEUC and SEUC. TIWs are mainly generated by shear instabilities between the nSEC and NECC (Philander, 1978; Athie and Marin, 2008) and between the EUC and the nSEC as well as baroclinic instability within the nSEC and cSEC (Weisberg and Weingartner, 1988; Jochum et al., 2004; von Schuckmann et al., 2008). Inter-simulation differences in the strength of the EUC, nSEC and NECC might generate differences in TIW activity. How the mesoscale dynamics impact the seasonal cycle of the off-equatorial subsurface currents is not clear and beyond the scope of the paper. However, comparing the seasonal cycle of TIW activity between the simulations might give additional insights into why we find different seasonal cycles for the NEUC but not for the SEUC in the two simulations. In general, we find a higher TIW activity in $CORE_{sim}$ compared to $JRA_{sim}$ (Fig. 7). Within the NEUC region, we find that the seasonal cycle of the TIW activity in $CORE_{sim}$ is dominated by an annual cycle and the seasonal maximum is nearly twice as high as in $JRA_{sim}$ (Fig. 7). The seasonal cycle of the TIW activity in $JRA_{sim}$ peaks in August and January and does not reveal a clear annual cycle. The

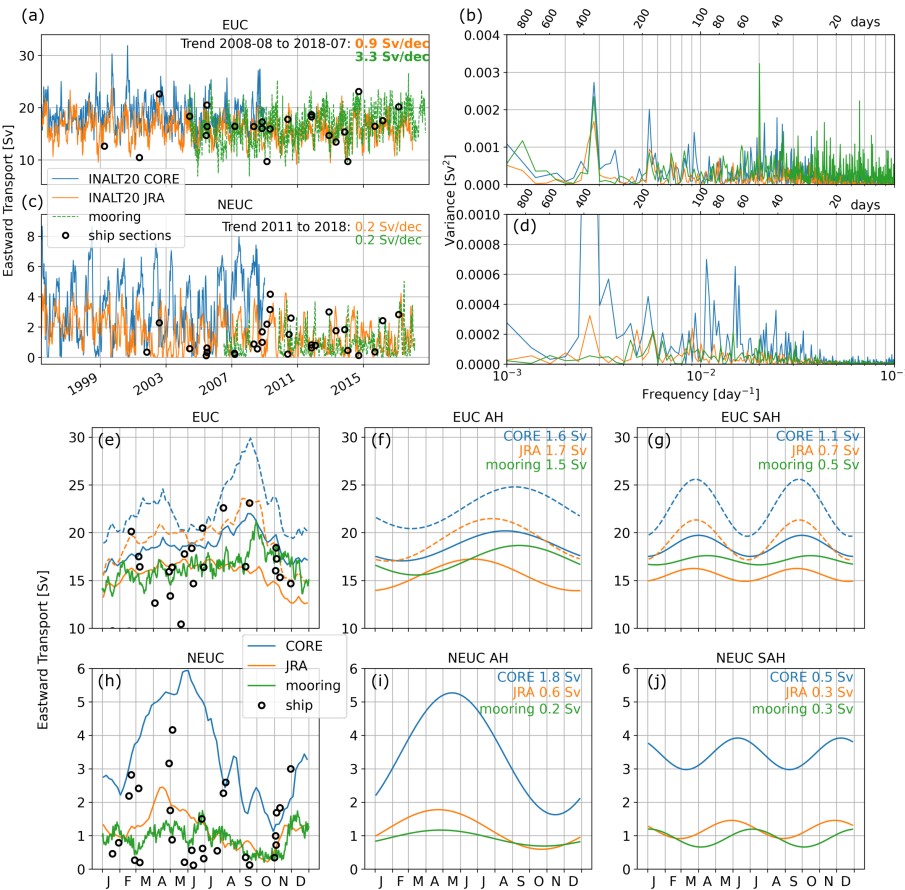

**Figure 5.** (a,c) Fixed box (EUC: 30 m-300 m, 1.2°S-1.2°N; NEUC: 60 m-270 m, 4.25°N-5.25°N) eastward transport time series (solid lines) and (b,d) power spectra of the EUC (a,b) and NEUC (c,d) calculated from CORE$_{sim}$ (EUC: Jun 1996-Dec 2009, NEUC: Nov 2001-Dec 2009, blue lines), JRA$_{sim}$ (EUC: Jun 2005-Dec 2018, NEUC: Nov 2010-Dec 2018, orange lines), moored observations (EUC: Jun 2005-Dec 2018, NEUC: Nov 2010-Dec 2018, green lines) and ship sections (black circles) at 23°W. (e,h) Seasonal cycle, (f,i) annual harmonic and (g,j) semi-annual harmonic of (e-g) EUC and (h-j) NEUC. Numbers in (f,g,i,j) represent the amplitude of the fitted harmonic cycle for each time series, respectively. The dashed lines in (e-g) show the results derived from eastward transports for the EUC calculated using Equ. 2 with a half mean width $W$ of 3° (CORE$_{sim}$: Jun 1996-Dec 2009, blue dashes lines, JRA$_{sim}$: Jun 2005-Dec 2018, orange dashes lines).

seasonal cycle of TIW activity in the SEUC region agrees well between the two simulations peaking in March and August. The differences in the seasonal cycle of mesoscale activity within the NEUC region between the two simulations might impact NEUC variability and hence contribute to the found inter-simulation discrepancies.

### 3.3   Long-term variability and trends

In this section we investigate the interannual and longer-term variability as well as linear trends of the wind field and current

transport in the simulations. Albeit longer-term variability and trends in the wind forcings are very uncertain, the wind field

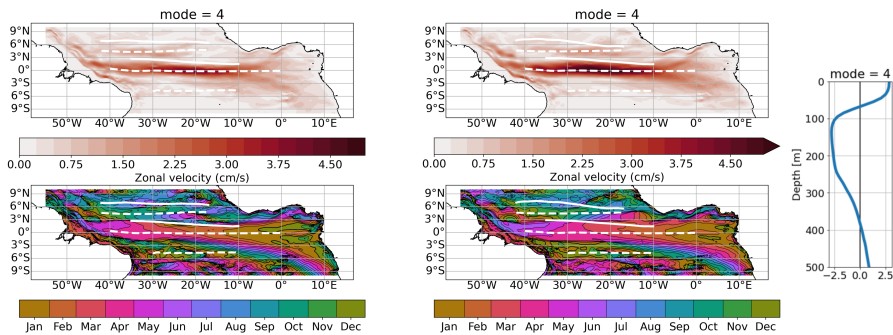

**Figure 6.** Amplitude (upper) and phase (lower) the fourth baroclinic mode, annual cycle of the zonal velocity from JRA$_{sim}$ (left, 2000-2018) and CORE$_{sim}$ (right, 1991-2009). To derive the 3D zonal velocity field associated with the specific baroclinic mode, the amplitudes must be multiplied by the corresponding vertical structure function shown on the right. The phase is given in month of the year when maximum eastward velocity occurs at the surface. Zonal white lines mark the mean latitude ($Y_{CM}$, Equ. 1) of the simulated surface (solid) and subsurface (dashed) currents for the respective periods.

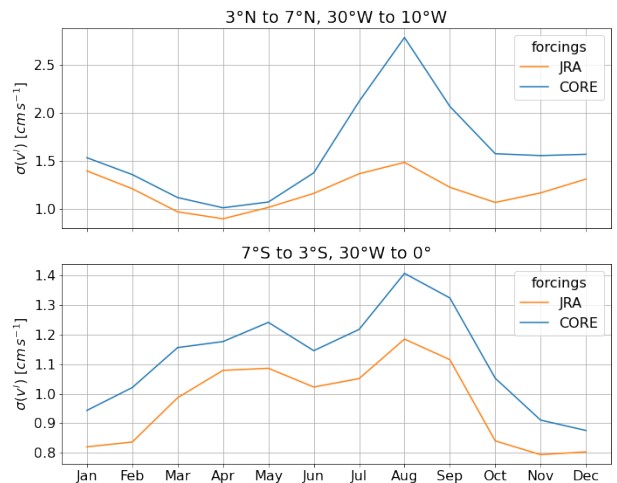

**Figure 7.** Monthly standard deviation of band-pass filtered meridional velocity at 160 m depth from JRA$_{sim}$ (orange line) and CORE$_{sim}$ (blue line) temporally averaged over the period 1980 to 2009 and spatially averaged within the NEUC (top) and SEUC region (bottom).

is expected to change under global warming. It is important to understand how longer-term changes and trends are related to changes in zonal currents. We start by briefly summarising the long-term variability and trends found by previous studies in the moored transport reconstructions of the EUC (Brandt et al., 2021) and NEUC (Burmeister et al., 2020) and briefly check if we can reproduce the results in JRA$_{sim}$ for an initial validation. This is not possible for CORE$_{sim}$ as it does not cover the time 485 period of observations. Linear trends are fitted to the time series within the given period from which the annual and semi-annual

harmonic were removed. Then we calculate the autocorrelation to find the degrees of freedom using the e-folding timescale. Finally, we test the significance of the trend using a two-sided t-test.

The 8-year moored transport time series of the NEUC is dominated by sporadic intraseasonal variability and does not reveal any longer-term variability or a linear trend between 2010 and 2018 (Fig. 5c,d; Burmeister et al., 2020). $JRA_{sim}$ realistically represent the result of the observations except for a small peak in the power spectra for the annual cycle which is not present in observations.

Brandt et al. (2021) observed that the EUC transport increased significantly by 3.3Sv/dec at 23°W between 2008 to 2018 (see also Fig. 5a). In $JRA_{sim}$, we find a significant but weaker increase of EUC transport (0.9Sv/dec) for the same period (Fig. 5a). Brandt et al. (2021) found that an intensification of trade winds in the western tropical North Atlantic and the concurrent strengthening of the STCs and enhanced Ekman divergence (1.1-2.0 Sv/dec depending on the wind product) can explain part of the observed EUC intensification. They suggested that the increase of the north-easterly trade winds might be associated with the Atlantic multidecadal variability (AMV, Delworth and Greatbatch, 2000), which switched from a warm phase in 2000s to a recent cold phase (Frajka-Williams et al., 2017). In $JRA_{sim}$ we find the Ekman divergence to significantly increase by 1.4 Sv/dec between 2008 and 2018 which agrees well with the results of Brandt et al. (2021). However, the increase of the EUC during this period in $JRA_{sim}$ is weaker.

The advantage of this study is that both model simulations go back to 1958 which enables us to compare the interannual to decadal variability of the wind forcings and the simulated zonal wind-driven current field. While the observational studies are not able to clearly identify if the linear trend found over the 10 years of observations is part of a longer-term variability or not, the longer time series from the simulations allow us to do so. Nevertheless, results especially with respect to decadal and longer variability must be regarded with great care as the forcing products are based on observations which span different time periods and fluctuate in their spatial coverage. Hence the decadal to longer-term variability in the simulations might not represent reality, especially in the earlier periods.

In the following, we removed the monthly mean seasonal cycle from 1980 to 2009 and averaged the simulated time series annually, which reveals the interannual to decadal variability. Please note, that in the following we use the path-following algorithm (Equ. 2) for the current transport of the individual zonal currents.

### 3.3.1 The wind field

The annual mean zonal wind stress anomalies in the $CORE_{sim}$ are stronger than those obtained for the $JRA_{sim}$, especially before 1970 (Fig. 8). For this early period, limited availability of observations on which the forcing products are based, might be one reason for the large inter-simulation discrepancies. While similarities between the two forcings exist in the western basin, differences increase toward the east of the basin. Largest differences between the two forcing products occur north of the equator before 1990.

To get a first impression how these spatial dissimilarities of the wind stress anomalies impact the zonal currents, we calculate the Sverdrup stream function (Equ. 10) using the annual mean wind stress curl anomalies, which we then average for different time periods (Fig. 9b-f). As a reference we calculate the Sverdrup stream function from the mean wind stress curl field from

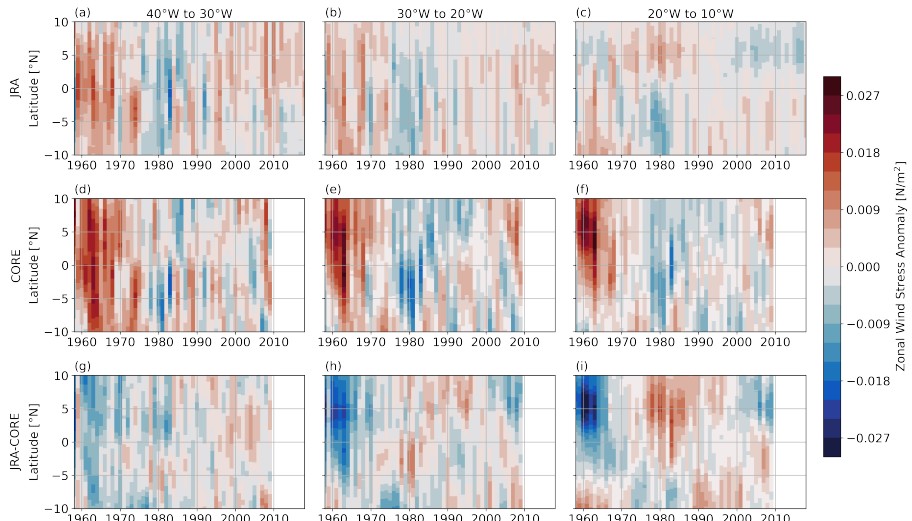

**Figure 8.** Hovmoeller diagram of annual mean zonal wind stress anomalies zonally averaged between $40°$W and $30°$W (left), $30°$W and $20°$W (centre) and $20°$W and $10°$W (right) for JRA$_{sim}$ (upper), CORE$_{sim}$ (middle), and the difference JRA$_{sim}$-CORE$_{sim}$ (lower). The anomalies are calculated by removing the seasonal cycle (1980-2009) from the monthly mean output before temporally averaging to annual resolution.

1980 to 2009 (Fig. 9a). For comparison with the model flow field, we also calculate anomalies of the stream function of the depth integrated velocity field of the upper 500 m (not shown) which compares well with the anomalous Sverdrup stream function indicating the importance of the Sverdrup stream function for interdecadal changes of the flow field. The spatial differences of the wind field anomalies result in distinct anomalous Sverdrup transports between the simulations, often with opposite sign for the shown periods. In the following, we present and discuss the longer-term variability and its connection to the wind field for each current separately. Therefore, we calculate the annual mean anomalous volume transport for each

current (Fig. 10). Additionally, we calculate the difference of the annual mean anomalous Sverdrup stream function between the approximate latitudinal boundaries (Equ. 11) of the currents at given longitudes (Fig. 11) assuming that the difference represents the zonal transport at that longitude (positive westward, negative eastward).

### 3.3.2   EUC

Before 1980, EUC transports are generally increasing in the two simulations (Fig. 10a-b). However, while in CORE$_{sim}$ lowest EUC transport anomalies (up to -6 Sv) across the entire time series occur before the mid-1970s, transport anomalies in JRA$_{sim}$ are still slightly positive during that period. Between 1980 to 2009, the EUC transport decreases in the two simulations (JRA$_{sim}$ -1.0 Sv/dec, CORE$_{sim}$ -0.4 Sv/dec) which is significant at a 95%-confidence level (Table 3). This is opposite to the increase of the EUC in the most recent decade (2008-2018) in observations and in JRA$_{sim}$ (Fig. 5 and Fig. 10a). Note that even though

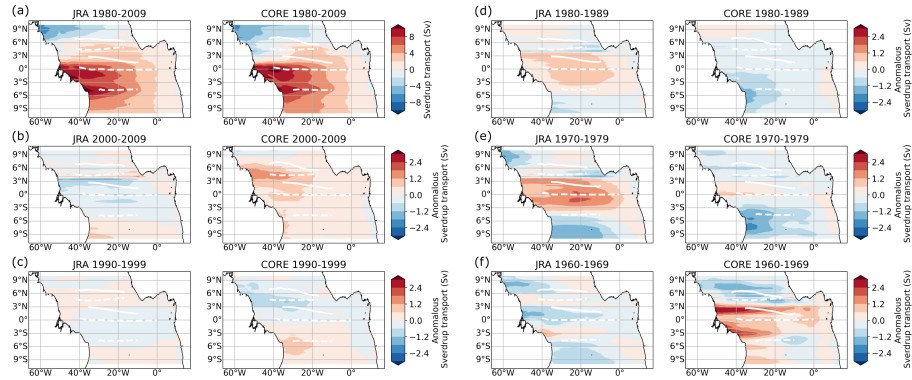

**Figure 9.** Sverdrup stream function calculated from (a) the 1980-2009 mean wind stress curl field, (b-f) annual mean wind stress curl anomalies averaged for the period (b) 2000-2009, (c) 1990-1999, (d) 1980-1989, (e) 1970-1979, (f) 1960-1969. In (b-f) we calculate first the Sverdrup stream function from the annual mean wind stress curl anomalies and average then over the respective periods. A negative stream function presents an anticlockwise rotation, this means that a zero-contour of the stream function with negative values in the south (north) marks maximum westward (eastward) velocities. Zonal white lines mark the mean latitude ($Y_{CM}$, Equ. 1) of the simulated surface (solid) and subsurface (dashed) currents for the respective periods.

we find a strengthening of the EUC in JRA$_{sim}$ in the last 10 model years, with respect to the 1980-2009 climatology, it is still anomalously weak.

Simultaneously to the EUC strengthening before the 1980s, easterly winds along the equator are intensifying in the two simulations with stronger westerly wind anomalies before 1970 in CORE$_{sim}$ compared to JRA$_{sim}$ (Fig. 8). This is accompanied by a positive trend in Ekman divergence of 3.6 Sv/dec in JRA$_{sim}$ and 3.9 Sv/dec in CORE$_{sim}$ between 1960 and 1980.

Likewise, the easterlies along the equator tend to decrease after mid-1980s and we find a negative trend as well in Ekman divergence between 1980 and 2009 of -1.5 Sv/dec in JRA$_{sim}$ and -0.9 Sv/dec in CORE$_{sim}$. Consequently, the EUC transport weakens during this period. Still, interannual anomalies of EUC transport differ between CORE$_{sim}$ and JRA$_{sim}$ which we link to the anomalous Sverdrup transport between 2°S and 2°N (Arhan et al., 2006). Before 1970, when westerly wind anomalies occur along the equator in the two simulations, we find anomalous westward Sverdrup transports in CORE$_{sim}$ (Fig. 11) which

is associated with a weakening of the EUC (Kessler et al., 2003; Arhan et al., 2006; Brandt et al., 2014). In contrast, the eastward Sverdrup transport anomalies in JRA$_{sim}$ along the equator resulting in positive EUC anomalies before mid-1970s. In the second period of anomalously weak easterlies along the equator from the early 1990s onward, the anomalous Sverdrup transport in CORE$_{sim}$ switches from eastward before 2000 to westward afterwards which impacts the EUC transports accordingly. In JRA$_{sim}$, the signal in the anomalous Sverdrup transport along the equator is less clear. However, after 2010,

the Ekman divergence acts to strengthen the EUC again while the easterly winds along the equator stay anomalously weak. The anomalous Sverdrup transport tends to be negative along the equator before mid-2010s which might counteract the EUC strengthening by the anomalous Ekman divergence.

Brandt et al. (2021) showed that the recent EUC strengthening is mainly related to trade wind changes in the western tropical North Atlantic (5°-10°N,60°-40°W) which result in the observed increased Ekman divergence in the tropical Atlantic. In agreement with Brandt et al. (2021), we find a switch from weaker north-easterly winds to stronger north-easterly winds in the western North Atlantic in the JRA$_{sim}$ from 2010 onwards. Due to its effect on the Northern Hemisphere trade winds, Brandt et al. (2021) suggested a link between EUC transport variability and the AMV. The AMV transitioned from a cold to warm phase from 1970 to 2010 and from a warm to cold phase before 1970 and after 2010. The north-easterly wind is weakest during a warm phase of the AMV. In general, our results support the idea that the decadal variability of the EUC is connected to the AMV through anomalous Ekman divergence which acts to strengthen (weaken) the EUC during a cold (warm) phase of the AMV.

### 3.3.3 NEUC

West of 30°W, NEUC transports in the two simulations decrease (switch from positive to negative anomalies) before 1980 and increase afterwards (Fig. 10c-d). At 35°W we find a significant trend of 0.6 Sv/dec in JRA$_{sim}$ and 0.8 Sv/dec in CORE$_{sim}$ between 1980 and 2009 (Tab. 3). While this signal is zonally coherent in CORE$_{sim}$, we find significant negative trends in current transport east of ∼30°W in JRA$_{sim}$. In the zonally averaged transports, we cannot find a significant trend in JRA$_{sim}$, while the NEUC transport is increasing by 0.8 Sv/dec in CORE$_{sim}$.

In JRA$_{sim}$ anomalies of zonal winds north of the equator are also not zonally coherent (Fig. 8). West of 20°W the easterlies north of the equator are strengthening before 1980, while they are weakening after 1980. In contrast east of 20°W the anomalies are reversed. Between 4°N and 6°N, the anomalous Sverdrup stream function drives eastward flow when the NEUC west of 30°W is anomalously strong, and westward flow when the western NEUC is anomalously weak (Fig. 11). The mean NEUC position west of 30°W is located along zero-contours of the anomalous Sverdrup stream function for all decades except for the 1990s (Fig. 9). East of 30°W the position of the NEUC is displaced northward of the zero-crossings which might explain why the NEUC anomalies are not zonally coherent in JRA$_{sim}$. In the CORE$_{sim}$, the NEUC anomalies are significantly correlated (R=0.75) with a strengthening/weakening of zonal easterly winds just north of the equator (2°-8°N,35°-15°W) between 1960 and 2009 (Fig. 12a). This is mainly associated with a switch of positive to negative zonal wind stress anomalies before 1980s. The correlation decreases for the period 1980 to 2009 (R=0.4).

Goes et al. (2013) suggested a link between the upwelling of the Guinea Dome region and the NEUC on interannual time scales. For the period 1960 to 2009, we find a significant correlation (JRA$_{sim}$ R=0.50, CORE$_{sim}$ R=0.47) between the box-averaged wind stress curl (5°-10°N, 35°-15°W) and the NEUC transport in the two simulations. When repeating the correlation for the period 1980 to 2009, it is still significant but decreases to R=0.39 in JRA$_{sim}$ while the correlation becomes non-significant in CORE$_{sim}$. In contrast to Goes et al. (2013), the results in JRA$_{sim}$ suggest a link between the NEUC and the upwelling within the Guinea Dome on interdecadal time scales. In CORE$_{sim}$ this link is less clear.

Tuchen et al. (2022b) recently reported decadal variability in TIW activity. As part of the NEUC is eddy driven (e.g. Jochum and Malanotte-Rizzoli, 2004; Assene et al., 2020), this might lead to long-term changes in the NEUC transports. However, we could not find a clear connection between long-term changes in TIW activity and NEUC transports (Fig. A4).

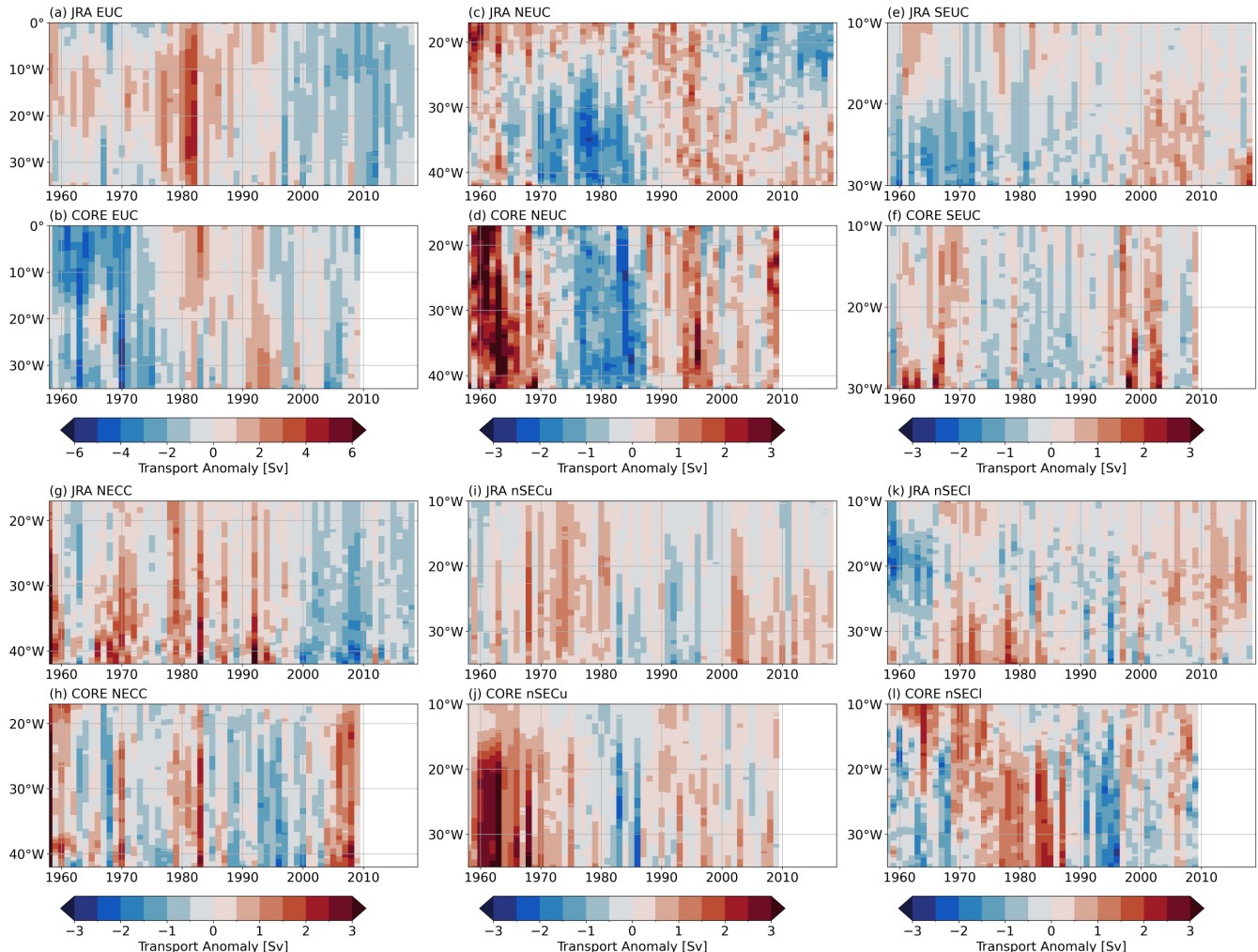

**Figure 10.** Hovmoeller diagram of annual mean transport anomaly (Sv) for (a,b) EUC (eastward current), (c,d) NEUC (eastward current), (e,f) SEUC (eastward current), (g,h) NECC (eastward current), (i,j) nSECu (westward current) and (k,l) nSECl (westward current). The anomalies are calculated by removing the seasonal cycle (1980-2009) from the monthly mean output before temporally averaging to annual resolution.

### 3.3.4 SEUC

Both model simulations show a significant increase in SEUC transports (JRA$_{sim}$ 0.4 Sv/dec, CORE$_{sim}$ 0.3 Sv/dec) between 1980-2009 (Table 3). Anomalies of the long-term variability of the SEUC also tend to be zonally coherent in the CORE$_{sim}$, while they can be of opposite sign east and west of about 20°W in the JRA$_{sim}$ (Fig. 10e-f). In both simulations, highest anomalies occur west of 20°W. In the JRA$_{sim}$, the SEUC west of 20°W shifts from a negative phase before mid-1990s to a positive phase afterwards. Likewise, the anomalous Sverdrup stream function acts to weaken (strengthen) the eastward flow of

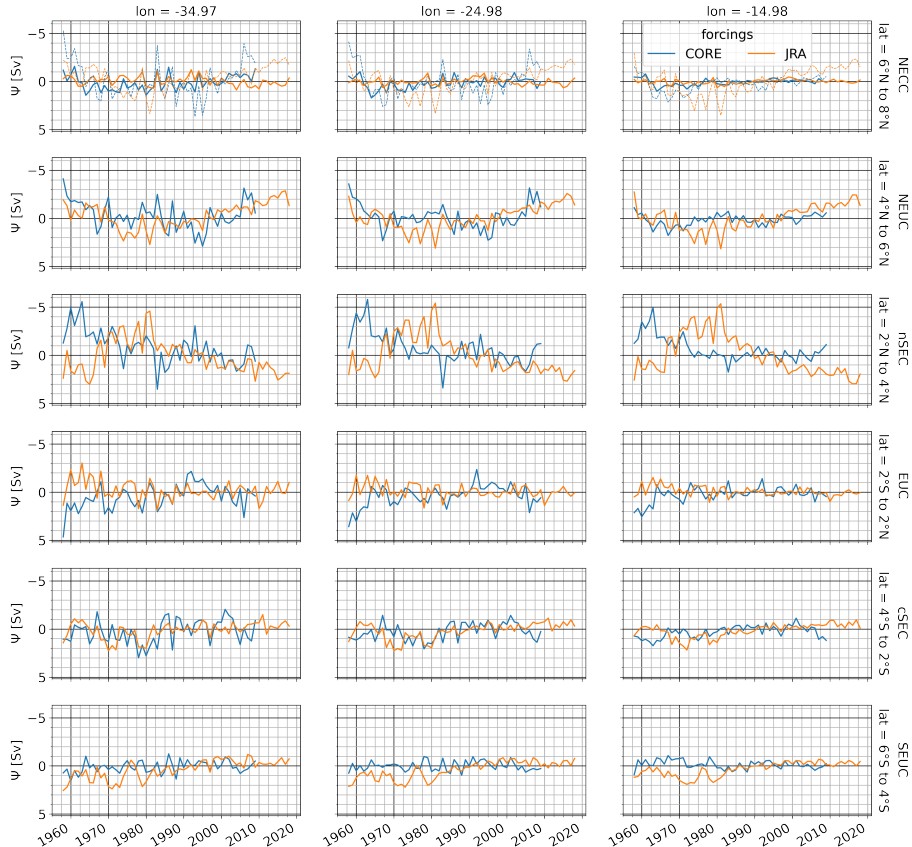

**Figure 11.** Difference of anomalous Sverdrup stream function Ψ with respect to 1980-2009 climatology calculated for different latitude bands centred above zonal current (rows, southern minus northern Ψ-value) for given longitudes (columns) for JRA$_{sim}$ (orange) and CORE$_{sim}$ (blue). Dashed lines in NECC row show the difference of Ψ between 4°N and 8°N as the NECC overlaps with the NEUC core position. Note that the y-axis is reversed as negative values indicate eastward flow anomalies.

the SEUC before (after) the 1990 (Fig. 11). East of 20°W the SEUC in the JRA$_{sim}$ varies by about 1-2 Sv on interannual to decadal time scales.

In the CORE$_{sim}$, the SEUC seems to covary with the NEUC on decadal time scales and we find anomalous negative (positive) wind stress curl averaged in a box south of the equator (10°S-0°, 35°-15°W) before the 1970s and after 1990s (between 1970 and 1990). The zonal flow associated with the anomalous Sverdrup stream function between 4°S and 6°S shows no clear link to the SEUC transport variability on decadal time scales but might explain some of the interannual variability (Fig. 11). The SEUC position in CORE seems to coincide with the maximum Sverdrup stream function which indicates a meridional

exchange with its flanking westward current bands of the cSEC (Fig. 9). Previous studies showed that the SEUC is mainly fed through recirculation with the ocean interior (Hüttl-Kabus and Böning, 2008; Fischer et al., 2008) and mesoscale eddy fluxes or mesoscale vortices are suggested to be one of the drivers of the SEUC (Jochum and Malanotte-Rizzoli, 2004; Assene et al.,

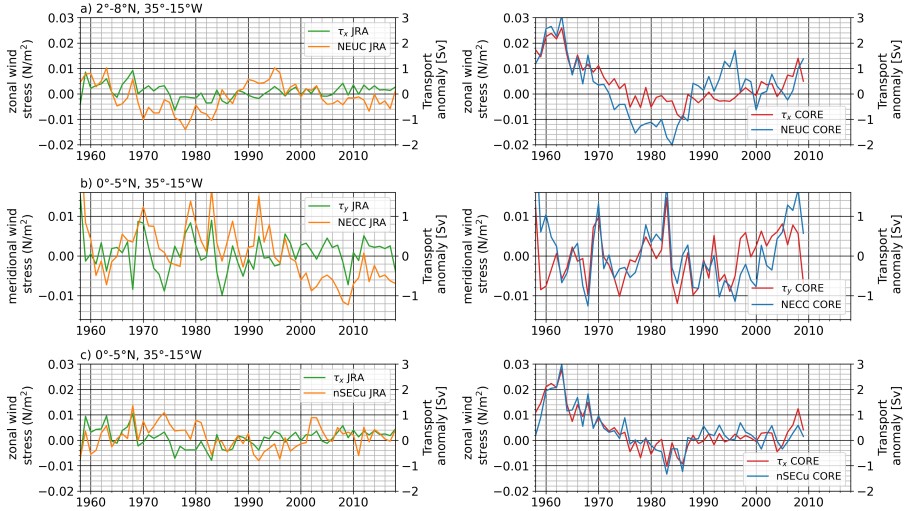

**Figure 12.** (a) Annual mean zonal wind stress anomalies averaged between $2°N$ and $8°N$, $35°W$ and $15°W$ (green and red lines) and zonally averaged annual mean NEUC transport anomalies (orange and blue lines) for $JRA_{sim}$ (left) and $CORE_{sim}$ (right). (b) Annual mean meridional wind stress anomalies averaged between $0°N$ and $5°N$, $35°W$ and $15°W$ (green and red lines) and zonally averaged annual mean NECC transport anomalies (orange and blue lines) for $JRA_{sim}$ (left) and $CORE_{sim}$ (right). (c) Annual mean zonal wind stress anomalies averaged between $0°N$ and $5°N$, $35°W$ and $15°W$ (green and red lines) and zonally averaged annual mean nSECu transport anomalies (orange and blue lines) for $JRA_{sim}$ (left) and $CORE_{sim}$ (right).

2020). As for the NEUC, however, we could not find a clear connection between long-term changes in TIW activity and SEUC transports (Fig. A4).

### 3.3.5 NECC

NECC transport anomalies tend to be zonally coherent in the two simulations (Fig. 10g-h), however after mid-1980s the anomalies are of different sign in $JRA_{sim}$ and $CORE_{sim}$. We find a decrease of -0.6 Sv/dec of the NECC transports in $JRA_{sim}$, and an increase of 0.2 Sv/dec in $CORE_{sim}$ between 1980 to 2009 (Table 3). The NECC anomalies after 1990 are associated with an anomalous Sverdrup stream function of opposite sign in the NECC region between $JRA_{sim}$ and $CORE_{sim}$ (Fig. 9b,c and 11). In $CORE_{sim}$, the zonal flow associated with the anomalous Sverdrup stream function between $4°$ and $8°N$ seem to better represent the long-term variability of the NECC while in $JRA_{sim}$ the anomalous Sverdrup stream function between $6°$ and $8°N$ seems to dominate flow variability.

Goes et al. (2013) and Hormann et al. (2012) suggested a link between the NECC variability and the Atlantic Meridional Mode, one of the dominant modes of Tropical Atlantic Variability which is acting on interannual to decadal time scales. The Atlantic Meridional Mode is characterised by a cross-equatorial sea surface temperature (SST) gradient and anomalous meridional winds blowing from the colder to the warmer hemisphere. It is mainly governed by the wind-evaporation-SST feedback (Carton et al., 1996; Chang et al., 1997). Goes et al. (2013) found a positive correlation between the NECC transports

and the meridional wind stress anomalies averaged in the box 0°-5°N, 35°-15°W just south of the NECC. We find a similar relationship on interannual time scales in the two simulations (Fig. 12b), despite the distinct inter-simulation discrepancies of the NECC on decadal time scales, especially after 2000.

### 3.3.6 nSEC (upper and lower)

In $CORE_{sim}$, the anomalies of the westward flowing nSECu on interannual to decadal time scales (Fig. 10i-j) are concurrent with anomalous easterlies just north of the equator (Fig. 8d-f). The nSECu and the easterlies are weaker before the mid-1970s, then they are stronger until the late 1980s. After 1990 they show weak positive anomalies (weakening) until the 2000s and then covary on interannual timescale at least between 30°-20°W. The correlation coefficient between annual anomalies of zonally averaged nSECu transports and box-averaged zonal wind stress (0°-5°N, 35°-15°W) is 0.90 at 0 lag in $CORE_{sim}$ (Fig. 12c). This is not the case in $JRA_{sim}$ where the correlation coefficient is 0.35 at lag 0. It seems that the zonal wind stress anomalies (Fig. 8a-c) in $JRA_{sim}$ lead the nSECu transport anomalies by 5-10 years. Indeed, we find maximum correlation (R=0.45) between annual anomalies of zonally averaged nSECu transports and box-averaged zonal wind stress (0°-5°N, 35°-15°W) with the wind stress leading 7 years. Still the correlation between nSECu transport and zonal wind stress is weak compared to CORE.

For the lower part of the nSEC (Fig. 10k-l), the two simulations tend to be in a better agreement regarding the long-term variability with stronger nSECl flow before 1970 and during late-1980s to late 1990s, and weaker flow between 1970-late 1980s and after late 1990s. However, in the $CORE_{sim}$ anomalies are stronger and seem to propagate from the eastern to the western basin while this is less clear in the $JRA_{sim}$. The anomalies of the nSECu and nSECl seem to be largely in phase in the $JRA_{sim}$, while they tend to vary out of phase in the $CORE_{sim}$. Interestingly in the $JRA_{sim}$ after 1980 between 30°W and 20°W, the nSEC and the NECC are the two strengthening or weakening at the same time, while this link is less clear in $JRA_{sim}$ before the 1980s and in the $CORE_{sim}$ for the entire period.

For the zonally averaged nSECu (surface), $JRA_{sim}$ simulates no significant trend, while $CORE_{sim}$ shows a significant decrease of 0.2 Sv/dec between 1980 to 2009. For the same period, the zonally averaged nSECl (subsurface) in $JRA_{sim}$ weakens by 0.1 Sv/dec, while transports strengthened by -0.2 Sv/dec in $CORE_{sim}$ (Table 3). Looking at trends of the two currents at selected longitudes reveals that they are not zonally coherent in the two simulations. For example, both simulations show a nSECl strengthening of 0.2 Sv at 15°W between 1980 and 2009, while at 35° the nSECl trend is negative and significant in $JRA_{sim}$ and negative but not significant in $CORE_{sim}$.

## 4 Summary and conclusion

In this study we investigate the effect of different wind forcings on the representation of zonal current strength and variability in the tropical Atlantic in a general ocean circulation model. The first forcing product is the CORE v2 dataset covering the period 1948 to 2009 (Griffies et al., 2009). It has a horizontal resolution of 2°x2° and temporal resolution of 6-hours. The second forcing product is the JRA55-do surface dataset (Tsujino et al., 2018). This dataset stands out due to its high horizontal ( 55

**Table 3.** Linear trends per decade of transports of the currents from 1980 to 2009. Significant trends in bold.

| Current | Lon | JRA$_{sim}$ (sign. 95%) | CORE$_{sim}$ (sign. 95%) |
|---|---|---|---|
| | | Sv/dec | Sv/dec |
| EUC | 35°W-0° | **-1.0** | **-0.4** |
| NEUC | 42°W-17°W | 0.1 | **0.7** |
| SEUC | 30°W-10°W | **0.4** | **0.3** |
| NECC | 42°W-17°W | **-0.6** | 0.2 |
| nSECu | 35°W-10°W | 0.1 | **0.2** |
| nSECl | 35°W-10°W | **0.1** | **-0.2** |
| EUC | 15°W | **-1.5** | **-0.8** |
| NEUC | 17°W | **-0.6** | 0.2 |
| SEUC | 15°W | -0.1 | 0.2 |
| NECC | 17°W | **-0.3** | **0.1** |
| nSECu | 15°W | -0.1 | 0.3 |
| nSECl | 15°W | **0.2** | **0.2** |
| EUC | 25°W | **-1.3** | **-0.3** |
| NEUC | 25°W | **-0.3** | **0.7** |
| SEUC | 25°W | **0.5** | **0.4** |
| NECC | 25°W | **-0.5** | **0.3** |
| nSECu | 25°W | **0.2** | 0.2 |
| nSECl | 25°W | **0.4** | -0.3 |
| EUC | 35°W | **-0.7** | **-0.4** |
| NEUC | 35°W | **0.6** | **0.8** |
| SEUC | 30°W | **0.6** | **0.6** |
| NECC | 35°W | **-0.8** | -0.1 |
| nSECu | 35°W | **0.4** | 0.5 |
| nSECl | 35°W | **-0.5** | -0.7 |

km) and temporal resolution (3 h) which covers the entire observational period (1958 to 2018). Where possible, we compare the results to ship sections and moored transport reconstructions along 23°W and 35°W (Brandt et al., 2021; Burmeister et al., 2020; Tuchen et al., 2022a).

The wind stress field of the CORE forcing is generally stronger than that of the JRA forcing on all timescale (Fig. 1, 3, 8), which is also reflected in the current transports except for the SEUC (Fig. 2, 4,5,10). In the mean fields between 1980 and 2009, JRA$_{sim}$ seems to better represent the EUC, NEUC, NECC and nSECl (Fig. 2). Depending on the individual currents,

the two simulations agree better in the western (NEUC, NECC, nSECl) or eastern basin (EUC, nSECu). The SEUC transports agree well between the two simulations which both underestimate the SEUC strength compared to observations. We find stronger positive wind stress curl in $CORE_{sim}$ at the western boundary as well as north of the equator in the central basin along 5°N, in the Guinea Dome region and along the coast of Northwest Africa (Fig. 1). South of the equator away from the western boundary the mean wind stress curl fields agree well. We find that part of the inter-simulation discrepancies can be explained by the coarser spatial resolution of the CORE forcing especially east of 30°W, north of the equator (Fig. 1 and A5). We also find higher wind driven upwelling in the Guinea Dome in $CORE_{sim}$ which can contribute to the higher NEUC transports compared to $JRA_{sim}$ (Stramma et al., 2005; Hüttl-Kabus and Böning, 2008; Goes et al., 2013). Stronger easterly winds along the equator can contribute to higher transports of the nSEC and EUC (Wacongne, 1989), which might be one reason for the higher mean current transports in $CORE_{sim}$ compared to $JRA_{sim}$. Additionally we find higher divergence of the meridional Ekman transport between 10°S and 10°N and higher zonal transport resulting from the meridional Sverdrup transport between 2°S and 2°N in $CORE_{sim}$ compared to $JRA_{sim}$, which again can contribute to inter-simulation differences of the EUC transports (Brandt et al., 2021; Arhan et al., 2006).

$CORE_{sim}$ generally features a stronger seasonal cycle of zonal wind stress and wind stress curl (Fig. 3) resulting in stronger seasonal cycle in current transports at 35°W and 23°W except for the SEUC (Fig. 4). The two simulations agree better in amplitude and phase of zonal wind stress and wind stress curl (Fig. 3) as well as current transport in the western basin than in the eastern basin (Fig. 4). To investigate the dynamical response of the zonal current field to the seasonal wind forcing we perform a baroclinic mode decomposition (Fig. 6 and A3). The phase speed of the first five baroclinic modes agree well between the two simulations (Tab. 2) which suggests that the differences in the seasonal cycle of current transport can be mainly attributed to differences in the wind forcings. We find a 2-3 month phase shift in the annual harmonic of EUC transport between the two simulations with $CORE_{sim}$ better representing the annual cycle found in observations. $JRA_{sim}$ realistically captures the sporadic seasonal fluctuations which dominate the NEUC transport variability in observations while the NEUC in $CORE_{sim}$ is unrealistically energetic on all time scales and is dominated by an overly strong seasonal cycle (Fig. 5). In contrast, the seasonal cycle of the SEUC transport is in good agreement between the two simulations. Differences of the annual cycle of the first two baroclinic modes between the two simulations may contribute to the discrepancies of the seasonal NEUC transports between $CORE_{sim}$ and $JRA_{sim}$ (Fig. A3). We also find different (similar) simulated seasonal TIW activity within the NEUC (SEUC) region between the two simulations (Fig. 7). As the NEUC and SEUC are thought to be partly eddy driven (e.g. Jochum and Malanotte-Rizzoli, 2004; Assene et al., 2020), this might be another reason for the discrepancies found in the simulated seasonal current transports. However further analysis is needed to confirm this which is beyond the scope of this paper.

On interannual to decadal time scales, $JRA_{sim}$ and $CORE_{sim}$ show opposite signs of annual mean zonal wind stress anomalies east of 20°W and north of the equator (Fig. 8g-i). The difference of the spatial pattern of wind field anomalies results in different anomalous Sverdrup flow (Fig. 9 and 11) which again can contribute to the differences in the long-term current variability between the two simulations. Interestingly, the anomalous stream function of the depth integrated velocity field in the upper 500 m (not shown) is similar to the anomalous Sverdrup flow on decadal time scales (Fig. 9) highlighting the importance

of decadal changes in the Sverdrup transport for the flow field in the tropical Atlantic. Between the two simulations, we find some similarities for the current strength of the NEUC and nSECl on interannual to longer-term timescales (Fig. 10c,d,k,l) while there is low agreement for EUC, SEUC, NECC, nSECu (Fig. 10a,b,e-j). For the EUC, we find that the anomalous Sverdrup transport between 2°S and 2°N can be one reason for inter-simulation differences of the transport variability on interannual time scales. Between 1960 and 2009, we find that the decadal variability of the NEUC is significantly correlated with the wind stress curl above the Guinea Dome region in the two simulations (JRA$_{sim}$ R=0.50, CORE$_{sim}$=0.47). This correlation however becomes non-significant in CORE$_{sim}$ when limiting the period to the last 30 years of the simulation (1980 to 2009). In JRA$_{sim}$ the longer-term variability of the SEUC seems to be associated with the anomalous Sverdrup stream function, while in CORE$_{sim}$ this link might explain some of the interannual transport variability, but it is less clear on decadal time scales (Fig. 11). Even though the NEUC and SEUC are partly eddy driven, we did not find a clear link between the long-term variability of TIWs and the strength of the off-equatorial subsurface currents (Fig. A4). While the nSEC in CORE$_{sim}$ shows high correlation (R=0.9) with the zonal wind stress just north of the equator on interannual to decadal time scales, the correlation is weaker (R=0.35) in JRA$_{sim}$. In the two simulations the NECC transports and the meridional wind stress anomalies just south of the NECC are related on interannual to decadal time scales (Fig. 12b), despite distinct differences of the longer-term current variability between the two simulations.

The JRA forcing is the successor of the CORE forcing for several ocean general circulation models. The application of the two different forcing products to a high-resolution ocean model, INALT20, provides us with two simulations resolving the complex zonal current field in the tropical Atlantic and allows us to compare the impact of different forcings on the ocean current field. Even though forced model simulations are needed to investigate the decadal and longer variability of ocean currents, it did not escape our notice that, without observations, we cannot validate which of the simulated decadal variability is more realistic. As the JRA forcing covers the modern period of observations and the period of the CORE forcing, JRA is forming a bridge to fill this knowledge gap. For example, Brandt et al. (2021) observed a strengthening of the EUC between 2008 and 2018, which we also found though weaker in JRA$_{sim}$. Looking at the entire simulation period, the two simulations suggest that the EUC transport is in a weak phase since the late 1990s and it is still recovering (Fig. 10). The model results indicate a decadal variability of the EUC which generally supports the assumption of Brandt et al. (2021) that the decadal EUC variability is linked to the AMV. Please note that this result needs to be regarded carefully, as one would need several 100-yr long integrations to make sound statements about multi-decadal variability like the AMV. Another example is that Goes et al. (2013) suggested a link between the NECC and meridional wind stress anomalies just south of the current, which are concomitant with the Atlantic Meridional Mode. Despite distinct inter-simulation discrepancies of the NECC long-term variability, the two model simulations support the link between the NECC strength and the meridional wind stress south of it on interannual to decadal time scales (Fig. 12b).

While it has become common for models to explain processes behind ocean observations, we postulate that velocity observations, once they have reached a critical length, need to be used to test the quality of wind-driven simulations. This paper presents one step in this direction. CORE and JRA are used in many published analyses. Here we have revealed some of their relative and absolute strengths and weaknesses simulating the upper ocean wind-driven circulation in the tropical Atlantic.

*Code and data availability.* All necessary code for the data analysis and preparation of the figures of this manuscript is freely available at https://github.com/Kristin-2002/Wind_forcing_public. All observational data supporting the findings of this study are publicly available as referenced within the paper. Model output necessary to reproduce the presented findings are available at https://data.geomar.de/downloads/20.500.12085/77c0d676-1933-4f17-9849-5ea2161736eb/.

**Appendix A**

**Table A1.** Meridional ship sections taken between 21°W and 28°W from 2000 to 2018. For all sections ADCP data is available. Sections including hydrography (CTD) measurements are marked accordingly. This data set is an extension of the data set used by Burmeister et al. (2020).

| cruise | date | averaged longitude | latitude | depth (m) | CTD |
|---|---|---|---|---|---|
| Meteor M47* | Mar-Apr 2000 | 23°W | 5°S-4°N | 500 | yes |
| Meteor M53 | May 2002 | 28°W | 5°S-2.5°N | 500 | yes |
| Meteor M55 | Oct-Nov 2002 | 24°W | 0°N-10°N | 500 | no |
| Sonne S170 | May 2003 | 28°W | 6°S-2.5°S | 800 | yes |
| Ronald H. Brown A16N | Jun-Aug 2003 | 26°W | 6°S-10°N | 400 | no |
| Polarstern ANTXXII/5 | Jun 2005 | 23°W | 6°S-14°N | 250 | no |
| Ronald H. Brown PNE6* | Jun 2006 | 23°W | 5°S-13.5°N | 800 | yes |
| Ronald H. Brown PNE6* | Jun-Jul 2006 | 23°W | 5°S-14°N | 800 | yes |
| Meteor M68/2* | Jun-Jul 2006 | 23°W | 4°S-14°N | 800 | yes |
| L'Atalante IFM-GEOMAR 4* | Feb 2008 | 23°W | 2°S-14°N | 350 | yes |
| L'Atalante IFM-GEOMAR 4 | Mar 2008 | 23°W | 2°S-14°N | 300 | no |
| Polarstern ANTXXV/5 | Apr-May 2009 | 23°W | 6°S-14°N | 250 | no |
| Ronald H. Brown PNE09* | Jul-Aug 2009 | 23°W | 0°N-14°N | 600 | no |
| Meteor M80/1* | Oct-Nov 2009 | 23°W | 6°S-14°N | 500 | yes |
| Polarstern ANTXXVI/1 | Oct-Nov 2009 | 23°W | 6°S-14°N | 250 | no |
| Meteor M81/1 | Feb-Mar 2010 | 21°W | 6°S-13°N | 1200 | no |
| Polarstern ANTXXVI/4 | Apr-May 2010 | 23°W | 5°S-13.5°N | 250 | no |
| Ronald H. Brown PNE10* | May 2010 | 23°W | 0°N-14°N | 650 | yes |
| Maria S. Merian MSM18/2* | May-Jun 2011 | 23°W | 0°N-14°N | 600 | no |
| Maria S. Merian MSM18/3 | Jun 2011 | 23°W | 4°N-14°N | 500 | yes |
| Ronald H. Brown PNE11 | Jul-Aug 2011 | 23°W | 0°N-14°N | 600 | no |
| Maria S. Merian MSM22* | Oct-Nov 2012 | 23°W | 6°S-8°N | 600 | yes |
| Maria S. Merian MSM22 | Oct-Nov 2012 | 23°W | 0°N-14°N | 600 | no |
| Ronald H. Brown PNE13a | Jan-Feb 2013 | 23°W | 0°N-14°N | 600 | no |
| Ronald H. Brown PNE13b* | Nov-Dec 2013 | 23°W | 6°S-14°N | 700 | yes |
| Meteor M106* | Apr-May 2014 | 23°W | 6°S-14°N | 500 | yes |
| Polarstern PS88.2* | Oct-Nov 2014 | 23°W | 2°S-14°N | 1200 | yes |
| Endeavor EN-550* | Jan 2015 | 23°W | 2°S-14°N | 700 | yes |
| Meteor M119* | Sep-Oct 2015 | 23°W | 5.5°S-14°N | 600 | yes |
| Meteor M130* | Aug-Oct 2016 | 23°W | 6°S-14°N | 600 | yes |
| Ronald H. Brown PNE17* | Feb-Mar 2017 | 23°W | 4°S-14°N | 700 | yes |
| Meteor M145* | Feb-Mar 2018 | 23°W | 6°S-14°N | 700 | yes |

*Cruises used to derive the buoyancy frequency profile at 23°W, 0°N

**Table A2.** Meridional ship sections taken at 35°W from 1990 to 2006. For all sections ADCP and hydrographic data is available. This data set is from Hormann and Brandt (2007)

| cruise | date | longitude | latitude | depth (m)* |
|---|---|---|---|---|
| Meteor M14/2 | Oct 1990 | 35°W | 5°S-2.5°N | full |
| Meteor M16/3 | Jun 1991 | 35°W | 5.5°S-2.5°N | full |
| Meteor M22/2 | Nov 1992 | 35°W | 5°S-4°N | full |
| L'Atalante - CITHER 1 | Feb 1993 | 35°W | 5°S-7.5°N | 600 |
| Meteor M27/3 | Mar 1994 | 35°W | 5°S-4.5°N | full |
| Le Noroit - ETAMBOT 1 | Sep 1995 | 35°W | 5°S-7.5°N | 200 |
| Edwin A. Link ETAMBOT 2 | Apr 1996 | 35°W | 4.5°S-7.5°N | full |
| La Thalassa - Equalant 99 | Aug 1999 | 35°W | 5°S-7 °N | full |
| Meteor M47/1 | Mar 2000 | 35°W | 5°S-5°N | full |
| Sonne S152 | Nov 2000 | 35°W | 5°S-9°N | full |
| Oceanus OC365/4 | Mar 2001 | 35°W | 1°S-7°N | full |
| Ron Brown 0201 | Feb 2002 | 35°W | 6°N-7°N | full |
| Meteor M53/2 | May 2002 | 35°W | 5.5°S-8°N | full |
| Sonne S171 | May 2003 | 35°W | 5.5°S-6.5 °N | full |
| Meteor M62/2 | Aug 2004 | 35°W | 5.5°S-5°N | full |
| Meteor M68/2 | Jun 2006 | 35°W | 5°S-5°N | full |

*Depths marked as 'full' span the entire water column.

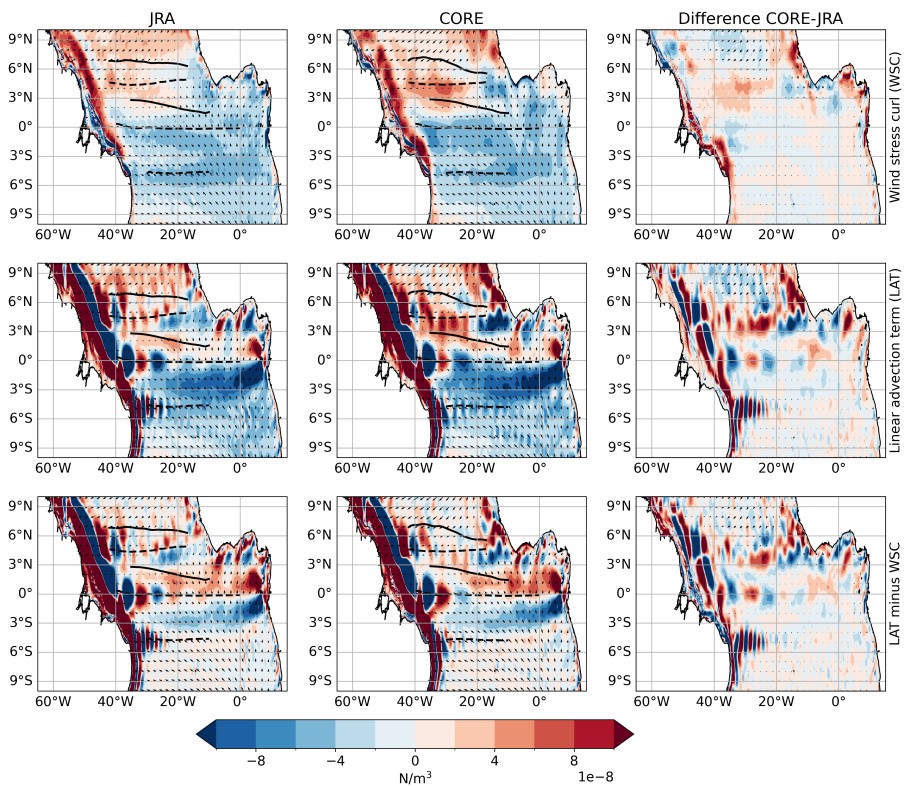

**Figure A1.** 1980 to 2009 mean maps of wind stress curl (WSC, colour shading, upper panel), wind stress (arrows), the linear advection term $\beta\rho_0 \int_{H=500\,m}^{0} v\,dz$ (LAT, colour shading, middle panel) and the difference of the two terms (colour shading, lower panel) calculated using JRA$_{sim}$ (left), CORE$_{sim}$ (centre), difference between the two forcings (right). Under Sverdrup balance LTA and WSC should be equal. Zonal black lines in mark the mean latitude ($Y_{CM}$, Equ. 1) of the simulated surface (solid) and subsurface (dashed) currents for the respective periods.

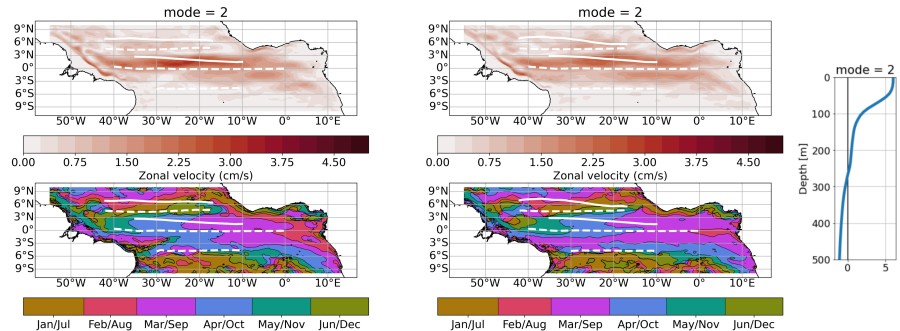

**Figure A2.** Amplitude (upper) and phase (lower) of second baroclinic mode, semi-annual cycle of zonal velocity from JRA$_{sim}$ (left, 2000-2018) and CORE$_{sim}$ (right, 1991-2009). To derive the 3D zonal velocity field associated with the specific baroclinic mode, the amplitudes must be multiplied by the corresponding vertical structure function shown on the right. The phase is given in month of the year when maximum eastward velocity occurs at the surface. Zonal white lines mark the mean latitude ($Y_{CM}$, Equ. 2) of the simulated surface (solid) and subsurface (dashed) currents for the respective periods.

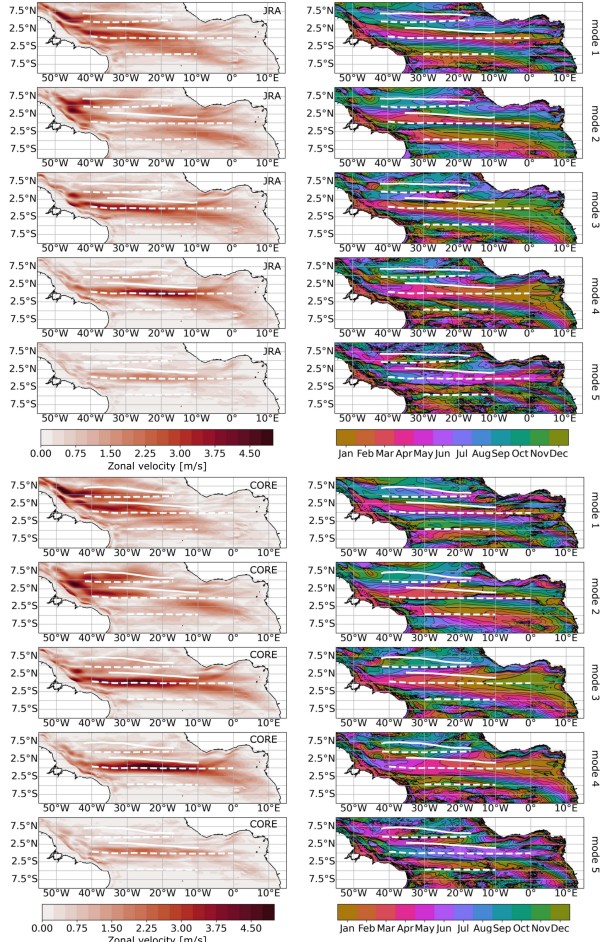

**Figure A3.** Amplitude (left) and phase (right) of first five baroclinic modes, annual cycle of zonal velocity from $\mathrm{JRA}_{sim}$ and $\mathrm{CORE}_{sim}$ (1980-2009). To derive the 3D zonal velocity field associated with the specific baroclinic mode, the amplitudes must be multiplied by the corresponding vertical structure function. The phase is given in month of the year when maximum eastward velocity occurs at the surface. Zonal white lines mark the mean latitude ($Y_{CM}$, Equ. 2) of the simulated surface (solid) and subsurface (dashed) currents for the respective periods.

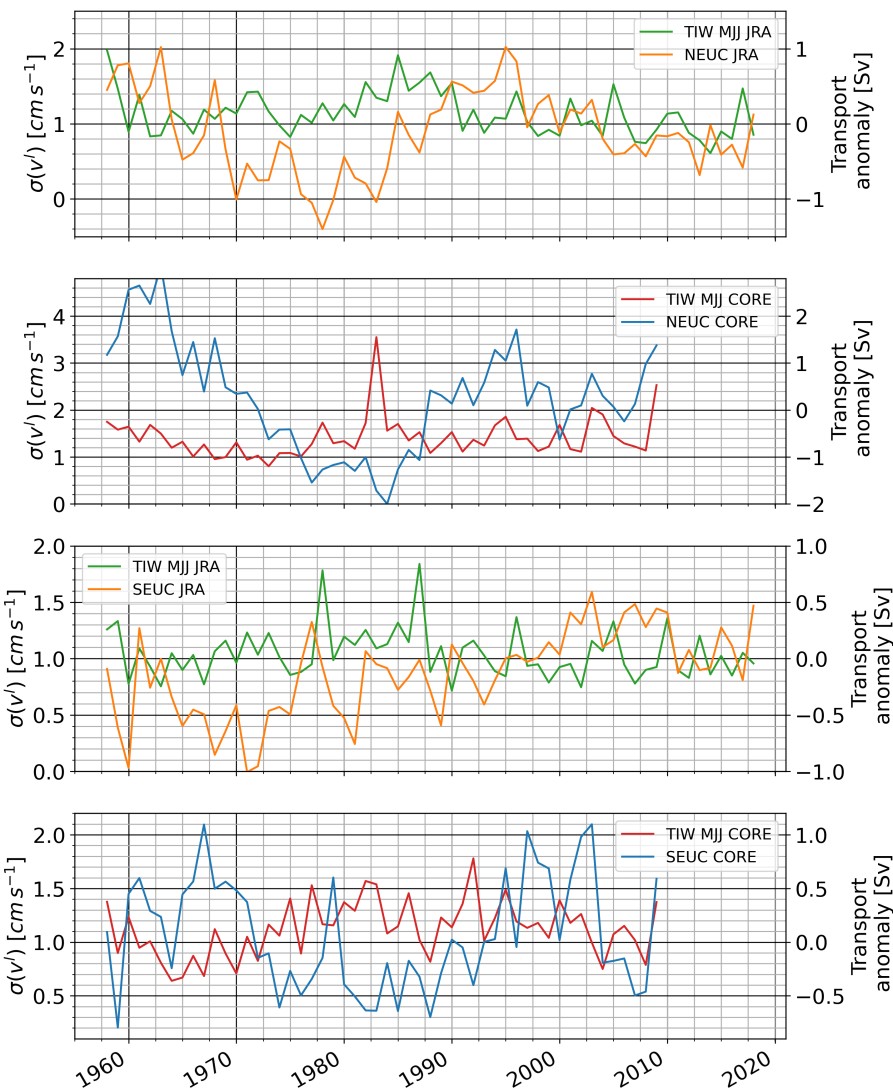

**Figure A4.** Long term TIW activity shown as May-June averages of monthly standard deviation of band-pass filtered meridional velocity at 160 m depth in JRA$_{sim}$ (green lines) and CORE$_{sim}$ (red lines) spatially averaged within the NEUC (top two panels) and SEUC region (bottom two panels). Also shown are the zonally averaged annual mean transport anomalies of the NEUC and SEUC in JRA$_{sim}$ (orange lines) and CORE$_{sim}$ (blue lines).

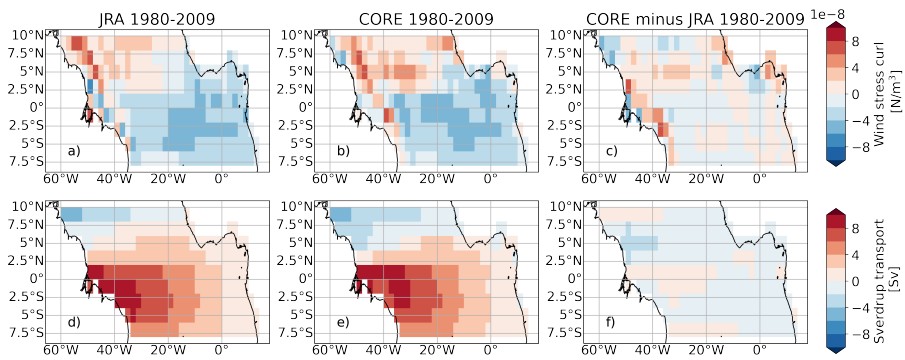

**Figure A5.** 1980 to 2009 mean maps of wind stress curl (a-c) and Sverdrup transport (d-f) calculated from wind stress data averaged in $2° \times 2°$ bins.

*Author contributions.* KB planned and performed the analysis, produced all figures and authored the manuscript from the first draft to the final version. FUS run the model simulations and advised in using the model output. WR provided the initial code to perform the modal decomposition of the model output. All co-authors contributed to the scientific improvement of the paper.

*Competing interests.* The contact author has declared that none of the authors has any competing interests.

*Acknowledgements.* This project has received funding from the European Union's Horizon 2020 research and innovation programme under grant agreement No 818123 (iAtlantic) and grant agreement No 817578 (TRIATLAS). This output reflects only the author's view, and the European Union cannot be held responsible for any use that may be made of the information contained therein. This study was funded by the Deutsche Forschungsgemeinschaft through several research cruises with RV L'Atalante, RV Maria S. Merian, RV Meteor, and RV Polarstern, by the project FOR1740 and by the Deutsche Bundesministerium für Bildung und Forschung (BMBF) as part of projects RACE (03F0651B), RACE-Synthesis (03F0824C) and SPACES-CASISAC (03F0796A). We thank the captains, crews, scientists, and technical groups involved in the different national and international research cruises to the eastern tropical North Atlantic that contributed to collecting CTD, velocity as well as mooring data, and making them freely available. Some of the velocity and oxygen observations were acquired within the PIRATA project and the CLIVAR TACE programme. The model integrations were enabled by the provision of computing resources on the HPC-systems JUWELS at the Jülich Supercomputing Centre (JSC) in the framework of the Earth System Modelling Project (ESM) and at the North German Supercomputing Alliance (HLRN). We thank one anonymous reviewer and Mike Bell for the constructive comments which greatly contributed to the improvement of the paper.

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
