# Peer review of "Dependency of simulated tropical Atlantic current variability on the wind forcing"

_EGUsphere, 2023_

## Referee Comment (RC2)

**Dependency of simulated tropical Atlantic current variability on the wind forcing**

**submitted to Ocean Sciences**

**Summary of paper**

This paper compares the zonal currents produced by two high resolution simulations of the equatorial Atlantic Ocean that differ only in their surface forcing (CORE and JRA) with in situ measurements from two series of cruises and two (sets of) moorings. The Sverdrup transports generated by the wind stresses are used to guide the comparisons. The CORE winds have stronger wind stress curl in the time-mean, particularly between 3°N and 6°N, and in their seasonal variation. Differences in depth-integrated transports and the NEUC are suggested to be largely caused by these differences. The SEUC is found to be weaker than the observations in both models. An interesting comparison is presented of variations in the currents at the two mooring sites, centred at (23°W, 0°N) and (23°W, 5°N). Seasonal variations in the currents are described and related to the "resonant" responses of some of the vertical normal modes. Variations in the under-currents on decadal time-scales are discussed, particularly the links between the AMV and EUC variations.

**Main comments**

The introduction gives a helpful overview of the context of the study, the observed circulation in the equatorial Atlantic and our understanding of it as a largely wind-driven system. The data and methods section generally provides good concise summaries. The results are presented in a clearly structured way.

My main scientific criticisms / suggestions are:

1) The paper relies very heavily on the depth integrated streamfunction calculated from the wind stress curl according to Sverdrup dynamics. This is used to try to infer information about the undercurrents and the surface currents. There are additional quantities related to the wind stresses which could be calculated to give additional information about the vertical structure of the flow. The Ekman transport would show the total wind-driven transport within the surface mixed layer and the Ekman upwelling at the base of the mixed layer given by curl ($\tau$ / f ) would give information about the vertical velocity at the base of the mixed layer that might be related to differences in the depths of the isopycnals in the two simulations. These expressions are not reliable very close to the equator, but are useful to within about 3° of it. The calculation of the meridional Ekman divergence described in lines 302-303 might also be applied nearer the equator, say at 3° rather than 10°, to obtain more information about Ekman upwelling near the equator. The authors have probably calculated these quantities already (as they have all the information required to do it). If they have not found them to be helpful in interpreting the results it would be helpful to explain that.

2) There is very little discussion of the meridional density structure and how much of the zonal current structure can be derived from it. Given the variability in the current data, that is very prominent in figures 4 and 5, the expectation that the density structure will be less "noisy", and the availability of this information from the models and most of the measurements, I would have thought that this would be a very useful "bridge" to make more sense of the results.

3) Given the emphasis on the Sverdrup streamfunction it might be good to compare it with the streamfunction of the depth integrated flow in the simulations themselves. Bottom pressure and other torques might cause them to differ; though I think bottom pressure torques near the equator are small.

I have to say that I found the paper quite a demanding read. This is partly because there are a lot of multi-part figures. It's partly because the relationships used to interpret the results are often rather qualitative; more than one relationship is often suggested as a possible explanation for a particular result. There were times, particularly on first reading, when I was not sure that I understood what was intended by a sentence or was confused about which method had been used to calculate a figure. So I have tried to make some suggestions that might help the reader grasp the paper more easily.

**Detailed comments**

**Abstract**

L7: "surface and subsurface"; it feels wrong to try to determine both just from one field (the wind stress curl). I think this will make many readers uncomfortable.

L9-10: "The simulated .. can, to a large extent, be explained …"  "To a large extent" seems an over-statement to me.

L10-12: Sentence on "recent strengthening of the EUC". I'm a bit concerned that the uncertainties in the decadal variations in the wind fields and how they drive the EUC and the small number of decades covered by the timeseries make this an over-statement as well. Some re-focusing of the sentence might allow something to be said that is substantive and more certain.

L 13 "postulate": this seems surprisingly tentative particularly on first reading.

**Introduction**

L17: ecosystems - would read better as "ecosystem"?

L26: It is assumed that the shallow overturning cells are wind-driven. I think this is true but references that demonstrate that would be helpful – or simply state that is assumed.

L38: As said already, the paper relies quite a lot on the Sverdrup streamfunction as a way of interpreting the current systems. It's rather like saying the Sverdrup streamfunction explains the thermal structure of subtropical gyres – they are largely wind-driven but ventilated thermocline type calculations are needed to understand the vertical structure. There is quite a lot of qualitative discussion; if one tried to make it more quantitative I suspect one would find the results rather unconvincing. This makes the paper difficult to read.

L49: "as a consequence": this is a very abbreviated explanation.

L60: Figure 1 appears a long way down the paper.

L62: I think NBUC is a typo (for NBC) and I could not find NBC (North Brazil Current) actually spelt out anywhere

L68: "NEC": I've lost track of this current. I think it is a westward surface current; it's not shown on Fig 1c.

L88: "the usage of … provides us with" : is unwieldy – re-write?

**Data and methods**

L94-101 This is a very informative summary but for some reason I did not recognise it as a summary of the section. Perhaps you could make that explicit and/or refer to subsections where the details are given.

L116-117: Are these simulations and their initial conditions identical except for the surface forcing? The radiative fluxes and precip also differ – not just the wind forcing.

L121: "well-established" here and in L246

L124: relatively

L133: "which uses satellite observations": this sounds a bit odd; surely they both assimilate satellite observations?

L136-137: A short summary of Table A1 would save most readers some time. E.g. most sections are at 23oW, cover 6S – 14N and the surface to 600 or 800 metres.

L145-146: "averaged … to derive mean sections" Is this time-averaging? Figures 2 and 4 obviously use different averaging.

L153: might help to say "zonal currents at a given longitude"

Equation 1: Subscripts u and l on $Z_u$ and $Z_l$ are lower case here and upper case in Table 1.

L162: "parameters" rather than "boundary conditions"?

L173: "full extent .. not always covered" It sounds like the coverage is not very good. This is disconcerting. One wonders whether it would be better to do comparisons by interpolating the model to the observations to calculate differences. Perhaps you could discuss or briefly mention somewhere the relative merits of extrapolating measurements to calculate standard transports versus interpolating models to the observation values and examining the differences.

L177-180: Does this regression estimate just the latitudinal variation (as a function of depth)

L178: replace 0° by 0°N

L181-187: Is there a scientific reason why different techniques are used for the two moorings?

L198- 201: As far as I can see the modal decomposition is only applied to model data in this study. So I'm puzzled why it is "important to note" this point.

Table 2: Is the CTD phase speed for mode 1 (2.51) a typo for 3.51?

L219: I think you mean a depth integral rather than a time integral

L221: projecting using a time-series that covers an integer number of years

Equation (11): this is the transport across a grid cell rather than the transport per unit length

**Results**

L259: "North of the equator". For consistency with previous sentence "North of these regions" would read more consistently.

L259: "Between 0°N and 5°N" might be better as "Between 2°N and 5°N"

L267: "equatorward" – I'm not sure what you have in mind here

L267: "mean position" helpful to add "near 6°N"

L271: I think Fig A5 and its discussion would go better here (before this new para) than in the summary.

L272-273: after "largest differences" insert "on this section". There are also large differences in the undercurrents around 400-500 m depth at 2.5S and 2.5N. The observed currents are stronger than the models'.

L273: It could be helpful to draw the 23 and 35W sections on Figure 1 (to see its meridional extent)

L274: extend -> extent

L275: "incoherence" I think should be "variation in the differences between"

L276: "current transport" please emphasise that these are the path following transports, referring to Table 1 or Eqn (2)

L281: The para starting here is quite difficult to read. It has a lot of qualitative arguments

Figure 1: it might be better for middle shade of wind stress curl to straddle zero? Best to define acronyms for currents once; either in this caption or perhaps in a Table.

Figure 2: The standard deviations aren't defined (e.g. std of annual means or monthly means) and are not mentioned in the text. Please state that the main contours in (a) – (f) show the observed values. It is odd that the meridional temperature/density structure is not highlighted more. In the equatorial Pacific at least it is very important.

L298: what do the +/- values indicate – the standard deviation of something?

L312: Sentence containing "eastern basin boundary". These tiny little features do not seem to contribute much to the difference field in Fig 1c. So do not seem very important; the lack of resolution of the CORE product at both boundaries seems more important. This sentence seems to interrupt (and confuse) the argument in the previous and following sentences.

L314: "can explain": you've given two other explanations so not clear why this one is emphasised.

L323: "current transports" say again that these are path-following ones.

L329: "Largest differences" in both wind stress and curl ?

L330: Perhaps say that elsewhere the agreement is quite good. The amplitudes of curl are small on equator; perhaps the wind stress magnitudes are large enough for the phase differences to be significant.

L336: underestimate – it would help the reader to refer to Fig 4g.

Fig 4a: The EUC obs seem to have a much smaller transport than the models at 23W. How does one reconcile this with fig 2 g.

Figure 4: The discussion of this is quite short (l 334-340). There is no discussion of the noisiness of the observations. This is quite disconcerting for the reader. Figure 5 suggests there is a lot of high frequency variability that the ship-based observations do not "resolve". This ought to be discussed. It might help to present Fig 5 before Fig 4?

L355: It's somewhat disturbing that changing W can give rise to a change in phase.

L 357: Judging from the spectra, the CORE model is more energetic in the intra-seasonal fluctuations. The large amplitude of the long period variations is perhaps making that less visible in the time-series.

Page 16: penultimate line: "in the equatorial Atlantic": is that just along the equator or at other latitudes too. Fig A2 shows that off the equator modes 1 and 2 are more important (as you say later).

L378: "about one month earlier". I find this quite hard to see and the contour spacing is 1 month so one needs to be careful.

L389: created -> create

L398: both -> the two ; differences in the TIW activity might be generated by the differences in the zonal current strengths (as well as the other way round)

L400: This summary para needs to be looked at again and re-worded. Are you only summarising differences between the two simulations?? "Both simulations" should say "The two simulations". The NEUC results in Figure 5 are worth highlighting?

L402: "as well as current transport" – I think seasonal variations in current data (figs 4 and 5) have only been presented at 23W and 35W ?

L 414: "significant trends". Section 3.3. on interannual variations talks about significant trends when most of them seem to be related to natural variability and the differences between the wind forcings are very large compared with the variation in the JRA wind anomalies. You need to explain in the methods section how significance is being assessed (and perhaps comment on what it means). In general I'm afraid I don't find the results in this section very convincing.

L428: It is very difficult to assess the reliability of decadal variations in the re-analysis wind products as the observation system on which they are based fluctuates and the number of decades that have been observed is very limited.

L437: I think these volume transports (Figure 10) are again defined using Eq (2) and Table 1. This should be made completely clear as we are being asked to compare with transport defined over the full depth in fig 11.

Fig 9 caption: The penultimate sentence should be included in the description of Fig 1 or in the text discussing Fig 1.

L447: anomalous -> anomalously ; do you mean compared to observations?

L474: "before 1980" but not before 1970 in CORE?

L480: "west of 20W" – do you mean east of 20W?

L481: What longitude range do you mean when you mention the western NEUC (e.g. west of 20W)?

L488: "wind stress curl south". This does not seem to be a typo for "north". I find the statement quite puzzling.

**Summary**

L559-560: If this is the case, why don't these wind stress differences also drive differences in the currents at 35oW (i.e. further to the west) ?

L 561-568: This para and Fig A5 is an important part of the discussion but it seems strange to discuss it for the first time in the summary. Could it not be discussed immediately after the discussion of Fig 1. Perhaps Fig 1 could be adjusted to contain Fig A5.

L 576-577: "As the subsurface off-equatorial …" The wind stress driving of the EUC is not much more direct is it? The sentence would be easier to read if it ended "than the EUC" (particularly as the EUC has not been discussed in the previous sentence or two).

L 581: "contribute" as mentioned earlier this could also be "partly due to"

L 588-589: "Most of the differences …" This is an important conclusion. I'm not really convinced that it is fully justified.

**Conclusions**

I'm not sure how the conclusions and the summary sections differ. Is the division really needed?

L 601-602: "The model results support …" This is also an important conclusion. But it is difficult to be sure about  multidecadal variability from such a short period. Perhaps there could be some discussion here of what simulations would be needed to obtain firm conclusions (e.g. 500 year simulations?)

L 606: It is strange to refer to Fig A4 for the first time here.

---

## Author Comment (AC1)

**Response to the reviews for**

**"Dependency of simulated tropical Atlantic current variability on the wind forcing"**

**by Kristin Burmeister, Franziska U. Schwarzkopf, Willi Rath, Arne Biastoch, Peter Brandt, Joke F. Lübbecke, and Mark Inall**

We thank all reviewers for the constructive comments. We edited the text accordingly. Additionally, we changed the colour of the ship sections in Fig.1 and added labels to Figure 8. Figure A1 in the appendix is new and was added to assess the applicability of the Sverdrup balance in the tropical Atlantic as a result of main comment three from Reviewer 2.

Detailed responses are found below. The comments by the reviewers are shown in black, our responses are given in red.

**Review of Burmeister et al.**

**Dependency of simulated tropical Atlantic current variability on the wind forcing**

**submitted to Ocean Sciences**

**Summary of paper**

This paper compares the zonal currents produced by two high resolution simulations of the equatorial Atlantic Ocean that differ only in their surface forcing (CORE and JRA) with in situ measurements from two series of cruises and two (sets of) moorings. The Sverdrup transports generated by the wind stresses are used to guide the comparisons. The CORE winds have stronger wind stress curl in the time-mean, particularly between 3° and 6°, and in their seasonal variation. Differences in depth-integrated transports and the NEUC are suggested to be largely caused by these differences. The SEUC is found to be weaker than the observations in both models. An interesting comparison is presented of variations in the currents at the two mooring sites, centred at (23°W, 0°N) and (23°W, 5°N). Seasonal variations in the currents are described and related to the "resonant" responses of some of the vertical normal modes. Variations in the under-currents on decadal time-scales are discussed, particularly the links between the AMV and EUC variations.

**Main comments**

The introduction gives a helpful overview of the context of the study, the observed circulation in the equatorial Atlantic and our understanding of it as a largely wind-driven system. The data and methods section generally provides good concise summaries. The results are presented in a clearly structured way.

My main scientific criticisms / suggestions are:

1) The paper relies very heavily on the depth integrated stream function calculated from the wind stress curl according to Sverdrup dynamics. This is used to try to infer information about the undercurrents and the surface currents. There are additional quantities related to the wind stresses which could be calculated to give additional information about the vertical structure of the flow. The Ekman transport would show the total wind-driven transport within the surface mixed layer and the Ekman upwelling at the base of the mixed layer given by curl $(\tau / f)$ would give information about the

vertical velocity at the base of the mixed layer that might be related to differences in the depths of the isopycnals in the two simulations. These expressions are not reliable very close to the equator, but are useful to within about 3o of it. The calculation of the meridional Ekman divergence described in lines 302-303 might also be applied nearer the equator, say at 3o rather than 10o, to obtain more information about Ekman upwelling near the equator. The authors have probably calculated these quantities already (as they have all the information required to do it). If they have not found them to be helpful in interpreting the results it would be helpful to explain that.

Thank you for this suggestion. Please find below figures of 1980-2009 mean (here Fig. 1) and seasonal averages (here Fig. 2) for the wind stress curl, the Sverdrup transport, the meridional Ekman transport, and Ekman pumping. We generally found that Ekman pumping and the meridional Ekman transport does not provide more information than wind stress curl or zonal wind stress. However, we provided key numbers of the Ekman transport at 10°N and 10°S the paper, to investigate the Ekman divergence along the equator. Please note, that we now also added values of Ekman pumping in the Guinea and Angola Dome which we use for the investigation of the NEUC and SEUC (L330-343).

[Figure]

*Figure 1: Mean maps of wind stress curl, Sverdrup stream function, meridional Ekman transport and Ekman pumping in JRA$_{sim}$ (left) and CORE$_{sim}$ (middle) and their difference (right) averaged from 1980 to 2009. Black lines mark the mean position of the currents in the simulations, respectively.*

[Figure]

*Figure 2: Difference (COREsim minus JRAsim) of seasonal averages of wind stress curl, Sverdrup stream function, meridional Ekman transport and Ekman pumping.*

2) There is very little discussion of the meridional density structure and how much of the zonal current structure can be derived from it. Given the variability in the current data, that is very prominent in figures 4 and 5, the expectation that the density structure will be less "noisy", and the availability of this information from the models and most of the measurements, I would have thought that this would be a very useful "bridge" to make more sense of the results.

Thank you for highlighting. Indeed, the zonal currents in the tropical Atlantic are mainly in geostrophic balance (e.g. Jochum and Malanotte-Rizzoli, 2004; Brandt et al., 2010; Goes et al. 2013). Thus, the density structure is similarly "noisy" than the zonal current structure.

3) Given the emphasis on the Sverdrup stream function it might be good to compare it with the stream function of the depth integrated flow in the simulations themselves. Bottom pressure and other torques might cause them to differ; though I think bottom pressure torques near the equator are small.

Thank you for this suggestion. Following Small et al. (2015), the depth integrated vorticity equation can be used to assess how much of the inter-simulation differences in the flow field can be attributed to the wind stress fields and the resulting Sverdrup transport. Under Sverdrup balance and to leading order, it can be expressed as the balance of the linear advection term $\beta\rho_0 \int_{-H}^{0} v \, dz$ and the wind stress curl, where $v$ is the simulated meridional velocity and $H = 500$ m is the depth of the active ocean layer of interest. Please note, the balance requires an integration depth where the vertical velocity is zero. Given that the isopycnals along that 500 m are fairly flat in the mean sections at 35°W and 23°W we assume that this criterion is fulfilled for long-term means. When subtracting the wind stress curl from the linear advection term the magnitudes in JRAsim and COREsim compare better in the central basin while differences remain in the spatial pattern (here Fig 3). The inter-simulation differences in the wind stress curl and the associated Sverdrup balance hence can explain only part of the difference found in the flow field north of the equator. We added the figure to the appendix of the manuscript (Fig. A1) and added a paragraph to the text (L319-329).

We also calculated decadal averages of anomalies of the stream function of the depth integrated meridional velocity in the upper 500m with respect to the 1980 to 2009 climatology and compared that to the Sverdrup stream function (Fig. 4). They compare well indicating the importance of the Sverdrup stream function for interdecadal changes of the flow field. We added two sentences to the text (L517-520).

[Figure]

*Figure 3 1980 to 2009 mean maps of wind stress curl (WSC, colour shading, upper panel), wind stress (arrows), the linear advection term $\beta\,\rho_0\,\int_{H=500m}^{0} v\ dz$ (LTA, colour shading, middle panel) and the difference of both (colour shading, lower panel)  calculated using  $JRA_{sim}$ (left), $CORE_{sim}$ (centre), the difference between both forcings (right). Under Sverdrup balance LTA and WSC should be equal. Zonal black lines mark the mean latitude of the simulated surface (solid) and subsurface (dashed) currents for the respective periods.*

[Figure]

*Figure 4: Decadal averages of the anomalous Sverdrup stream function and the anomalous stream function of the depth integrated meridional velocity field of the upper 500m.*

I have to say that I found the paper quite a demanding read. This is partly because there are a lot of multi-part figures. It's partly because the relationships used to interpret the results are often rather qualitative; more than one relationship is often suggested as a possible explanation for a particular result. There were times, particularly on first reading, when I was not sure that I understood what was intended by a sentence or was confused about which method had been used to calculate a figure. So I have tried to make some suggestions that might help the reader grasp the paper more easily.

Thank you for the extensive list of detailed comments to improve the clarity of the paper. We really appreciate the effort by the reviewer and took great care to implement the suggestions (see detailed responses to the individual comments below).

**Detailed comments**

**Abstract**

L7: "surface and subsurface"; it feels wrong to try to determine both just from one field (the wind stress curl). I think this will make many readers uncomfortable.

L9-10: "The simulated .. can, to a large extent, be explained …" "To a large extent" seems an overstatement to me.

L10-12: Sentence on "recent strengthening of the EUC". I'm a bit concerned that the uncertainties in the decadal variations in the wind fields and how they drive the EUC and the small number of decades covered by the timeseries make this an over-statement as well. Some re-focusing of the sentence might allow something to be said that is substantive and more certain.

L 13 "postulate": this seems surprisingly tentative particularly on first reading.

Thank you for the suggestions regarding the abstract. We edited the complete abstract to be more specific and avoid over-statements. (L1-22)

**Introduction**

L17: ecosystems - would read better as "ecosystem"?

Changed. (L24)

L26: It is assumed that the shallow overturning cells are wind-driven. I think this is true but references that demonstrate that would be helpful – or simply state that is assumed.

Apologies for not being clear enough. The "wind-driven" in the sentence relates only to the "wind-driven gyre circulation". We rearrange the order to avoid confusion (L35). We describe the different overturning cells and their drivers in the following text of the sentence with appropriate references.

L38: As said already, the paper relies quite a lot on the Sverdrup stream function as a way of interpreting the current systems. It's rather like saying the Sverdrup stream function explains the thermal structure of subtropical gyres – they are largely wind-driven but ventilated thermocline type calculations are needed to understand the vertical structure. There is quite a lot of qualitative discussion; if one tried to make it more quantitative I suspect one would find the results rather unconvincing. This makes the paper difficult to read.

Thank you for highlighting. We edited the manuscript throughout to tone down this statement and mention other dynamics which might play a role and were not analysed in this paper.

L49: "as a consequence": this is a very abbreviated explanation.

Thank you for highlighting. We edited the sentence for more clarity. (L52)

L60: Figure 1 appears a long way down the paper.

Changed.

L62: I think NBUC is a typo (for NBC) and I could not find NBC (North Brazil Current) actually spelt out anywhere

Thank you for highlighting. We spelt the currents out (L79-80). The source for the NBC is the NBUC (North Brazil Undercurrent), which is fed again by the SEC (e.g. Fig. 3 in Schott et al. 2004 and corresponding text). We added a sentence to explain this connection (L80-82).

L68: "NEC": I've lost track of this current. I think it is a westward surface current; it's not shown on Fig 1c.

Thank you for highlighting. The North Equatorial Current (NEC) is a westward current and part of the subtropical gyre in the North Atlantic. We included a brief explanation in the text (L58-59).

L88: "the usage of … provides us with" : is unwieldy – re-write?

Thanks for highlighting. We edited the sentence accordingly. (L104-105)

**Data and methods**

L94-101 This is a very informative summary but for some reason I did not recognise it as a summary of the section. Perhaps you could make that explicit and/or refer to subsections where the details are given.

Thank you for highlighting. We changed the paragraph accordingly. (L113-123)

L116-117: Are these simulations and their initial conditions identical except for the surface forcing? The radiative fluxes and precip also differ – not just the wind forcing.

Thank you for highlighting. The forcing products differ not only in the wind forcing. We edited the sentence accordingly. (L138)

L121: "well-established" here and in L246

Changed. (L143, L280)

L124: relatively

Changed. (L147)

L133: "which uses satellite observations": this sounds a bit odd; surely they both assimilate satellite observations?

Apologies for not being clear enough. CORE v2 uses Satellite-based radiation data and blended satellite-gauge precipitation data in place of the reanalysis data for 1984-2009 and 1979-2009, respectively. Prior to the satellite era, COREv2 does not contain realistic time-varying radiation and precipitation fluxes as climatological values are used to fill in missing years (Large and Yaeger, 2009). We added this information. (L143-146)

L136-137: A short summary of Table A1 would save most readers some time. E.g. most sections are at 23°W, cover 6°S – 14°N and the surface to 600 or 800 metres.

Thank you for this suggestion. We changed the text accordingly. (L162-163)

L145-146: "averaged … to derive mean sections" Is this time-averaging? Figures 2 and 4 obviously use different averaging.

Thank you for highlighting. We edited the sentence accordingly. (L171)

L153: might help to say "zonal currents at a given longitude"

Changed. (L178)

Equation 1: Subscripts u and l on Ztiu and Ztil are lower case here and upper case in Table 1.

Changed.

L162: "parameters" rather than "boundary conditions"?

Changed. (L189)

L173: "full extent .. not always covered" It sounds like the coverage is not very good. This is disconcerting. One wonders whether it would be better to do comparisons by interpolating the model to the observations to calculate differences. Perhaps you could discuss or briefly mention somewhere the relative merits of extrapolating measurements to calculate standard transports versus interpolating models to the observation values and examining the differences.

Thank you for highlighting. We edited the sentence for clarity (L209-215). Please note that the NEUC transport timeseries is derived and fully assessed in Burmeister et al. (2020) and was found to be a valid method to reconstruct the NEUC transport. Indeed between 2011 and 2016 the upper 10m of

the NEUC are not covered by the ADCP measurements which might lead to small but neglectable underestimation of the current. The vertical grid of the model cannot resolve at such a small scale. In the model the first vertical grid point below 65m is centered at 78m. Hence the comparison between model and observation seems appropriate.

L177-180: Does this regression estimate just the latitudinal variation (as a function of depth)

For this regression Hilbert empirical orthogonal functions (HEOFs) are calculated from the meridional ship sections. This method can detect moving features in space, like e.g. a meridional or vertical migration of the EUC. The velocity field is then reconstructed by regressing the composite pattern of the first three HOEFs on the moored observations. A detailed description can be found in Brandt et al. (2014)

L178: replace 0° by 0°N

Changed throughout the manuscript.

L181-187: Is there a scientific reason why different techniques are used for the two moorings?

Yes, while the variability of the EUC is characterised by meridional and vertical migration, it was found that the main mode of variability for the NEUC is a homogenous strengthening within the entire integration box and therefore the simpler optimal width methods is sufficient to capture the NEUC variability (see S3-S4 of Burmeister et al., 2020).

L198- 201: As far as I can see the modal decomposition is only applied to model data in this study. So I'm puzzled why it is "important to note" this point.

Thank you for highlighting. The vertical structure functions used for the modal decomposition of the model output are derived from the observations following Brandt et al. (2016). We describe this in the first sentence of the paragraph.

Table 2: Is the CTD phase speed for mode 1 (2.51) a typo for 3.51?

Thank you for highlighting. We found an error in our calculation. The model output was not interpolated to a sufficient vertical resolution before calculating $N^2$ which led to an erroneous mode decomposition. We corrected the text (L231-232) and table 2 accordingly (here table 1).

*Table 1: Result for gravity waves speeds in m/s for first five baroclinic modes of the gravest basin mode derived from the corrected $N^2$ profiles. Model and observations agree well with each other for all modes.*

|       | Mode 1 | Mode 2 | Mode 3 | Mode 4 | Mode 5 |
|-------|--------|--------|--------|--------|--------|
| Obs   | 2.51   | 1.40   | 0.98   | 0.76   | 0.57   |
| JRA   | 2.53   | 1.43   | 1.05   | 0.81   | 0.58   |
| CORE  | 2.51   | 1.42   | 1.04   | 0.80   | 0.57   |

L219: I think you mean a depth integral rather than a time integral

Changed. (L247)

L221: projecting using a time-series that covers an integer number of years

Changed. (L249)

Equation (11): this is the transport across a grid cell rather than the transport per unit length

We are afraid we do not know what the reviewer would like us to change.

**Results**

L259: "North of the equator". For consistency with previous sentence "North of these regions" would read more consistently.

Changed. (L293)

L259: "Between 0°N and 5°N" might be better as "Between 2°N and 5°N"

Changed. (L294)

L267: "equatorward" – I'm not sure what you have in mind here

"…to the south." Edited the sentence accordingly. (L299-300)

L267: "mean position" helpful to add "near 6°N"

Added. (L301)

L271: I think Fig A5 and its discussion would go better here (before this new para) than in the summary.

Thank you for this suggestion. We moved the paragraph about the resolution to the mean section. (L360-367)

L272-273: after "largest differences" insert "on this section".

Changed. (L309)

There are also large differences in the undercurrents around 400-500 m depth at 2.5S and 2.5N. The observed currents are stronger than the models'.

Thank you for highlighting. This is indeed true. This is the westward flowing equatorial intermediate current along the equator and the eastward flowing, off-equatorial Northern and Southern Intermediate Countercurrents (e.g. Brandt et al., 2015; Ascani et al., 2010; Ollitrault et al., 2006). Most ocean general circulation models struggle to reproduce the strength of the equatorial intermediate currents due to yet unidentified deficiencies in model dynamics (Ascani et al., 2010). Claus et al. (2016) studied the deep equatorial jets and suggests the vertical energy flux is important to maintain the jets locally which suggests the importance of the vertical resolution in simulations to realistically simulate energy fluxes. However, the results from Ascani et al. (2010) suggest that resolution alone is not the key.

L273: It could be helpful to draw the 23 and 35W sections on Figure 1 (to see its meridional extent)

Thank you for the suggestion. We indicated the 23° and 35°W section in Fig. 1a and b as white dashed line. We acknowledge that those are hard to see and added black lines to Fig. 1c.

L274: extend -> extent

Changed. (L310)

L275: "incoherence" I think should be "variation in the differences between"

Changed. (L312)

L276: "current transport" please emphasise that these are the path following transports, referring to Table 1 or Eqn (2)

Added. (L313)

L281: The para starting here is quite difficult to read. It has a lot of qualitative arguments

Thank you for highlighting. We edited the paragraph accordingly. (L330-343)

Figure 1: it might be better for middle shade of wind stress curl to straddle zero? Best to define acronyms for currents once; either in this caption or perhaps in a Table.

Thank you for the suggestions. We edited the caption to include all acronyms for the currents. For the colour shading we decided to keep it as it is for the following reasons:

1. to be consistent throughout the paper

2. to avoid mixing up areas of NaNs which are white (more relevant in other figures)
3. we think that the very pale red and blue colours around zero can be clearly identified to be close to zero.

Figure 2: The standard deviations aren't defined (e.g. std of annual means or monthly means) and are not mentioned in the text. Please state that the main contours in (a) – (f) show the observed values.

Thank you for highlighting, we edited the caption accordingly.

It is odd that the meridional temperature/density structure is not highlighted more. In the equatorial Pacific at least it is very important.

Thanks for highlighting. It is true, the equatorial currents are known to be in geostrophic balance. This means that stronger currents are associated with steeper sloping of isopycnals and vice versa. We added a few sentences to highlight this in the text (L 306-308)

L298: what do the +/- values indicate – the standard deviation of something?

Yes, thank you for highlighting. We edited the sentence for more clarity. (L370-372)

L312: Sentence containing "eastern basin boundary". These tiny little features do not seem to contribute much to the difference field in Fig 1c. So do not seem very important; the lack of resolution of the CORE product at both boundaries seems more important. This sentence seems to interrupt (and confuse) the argument in the previous and following sentences.

Thank you for highlighting. We edited the paragraph for clarity and moved it to the summary section to reduce the length of the paper. (L649-663)

L314: "can explain": you've given two other explanations so not clear why this one is emphasised.

Thank you for highlighting. We edited the paragraph for clarity and moved it to the summary section to reduce the length of the paper. (L649-663)

L323: "current transports" say again that these are path-following ones.

Added. (L389)

L329: "Largest differences" in both wind stress and curl ?

Yes, edited sentence for clarity. (L396-397)

L330: Perhaps say that elsewhere the agreement is quite good. The amplitudes of curl are small on equator; perhaps the wind stress magnitudes are large enough for the phase differences to be significant.

Thank you for highlighting. We added some sentences for clarity. (L397-403)

L336: underestimate – it would help the reader to refer to Fig 4g.

Added. (L405)

Fig 4a: The EUC obs seem to have a much smaller transport than the models at 23W. How does one reconcile this with fig 2 g.

The observations shown in Fig. 4 are individual ship sections. They are snapshots representing mainly short-term variability. This short-term variability is not reflected in a seasonal cycle averaged over a 30-year period. If compared to the time series (Fig. 5) the ship sections realistically reflect the high short-term variability of the EUC and fit well with model and observations as reflected by the mean and the standard deviation in Fig. 2g. For clarity we decided against showing the standard deviation in Fig. 4 (left column) and 5e,h.

Figure 4: The discussion of this is quite short (l 334-340). There is no discussion of the noisiness of the observations. This is quite disconcerting for the reader. Figure 5 suggests there is a lot of high

frequency variability that the ship-based observations do not "resolve". This ought to be discussed. It might help to present Fig 5 before Fig 4?

See answer above.

L355: It's somewhat disturbing that changing W can give rise to a change in phase.

That $JRA_{sim}$ does not simulate the seasonal cycle of the EUC and that the W needs to be increased to capture the seasonal cycle correctly is indeed concerning. Especially because the EUC is mainly driven by zonal wind stress along the Equator and its seasonal variations are due to resonant equatorial basin modes. We think this is mainly due to the spatial pattern of the seasonal cycle in the wind stress.

L 357: Judging from the spectra, the CORE model is more energetic in the intra-seasonal fluctuations. The large amplitude of the long period variations is perhaps making that less visible in the timeseries.

Thank you for highlighting. We edited the sentence accordingly. (L430-431)

Page 16: penultimate line: "in the equatorial Atlantic": is that just along the equator or at other latitudes too. Fig A2 shows that off the equator modes 1 and 2 are more important (as you say later).

Thank you for highlighting. We accidently took over the languages of the reference paper. In this case it means along the equator. We edited the sentence accordingly. (L441)

L378: "about one month earlier". I find this quite hard to see and the contour spacing is 1 month so one needs to be careful.

Thank you for highlighting. We calculated the averaged difference of the phase of the annual cycle in both simulations. Between 0° and 40°W along the equator (±0.5°), maximum velocities in the $JRA_{sim}$ occur on average 22 days earlier than in the $CORE_{sim}$ with a standard deviation of 6 days. We edited the text for clarity. (L450-452)

L389: created -> create

Changed. (L462)

L398: both -> the two ; differences in the TIW activity might be generated by the differences in the zonal current strengths (as well as the other way round)

Thank you for highlighting. We edited the sentence accordingly (L469). Please note, TIWs are mainly generated by shear instabilities between the nSEC and NECC and between the EUC and the nSEC as well as baroclinic instability within the nSEC and cSEC. Inter-simulation differences in the strength of the EUC, nSEC and NECC might generate differences in the TIW activity. We think it is unlikely that differences in the strength of the NEUC and SEUC can cause differences in TIW activities. We added a few sentence to the text. (L463-466)

L400: This summary para needs to be looked at again and re-worded. Are you only summarizing differences between the two simulations?? The NEUC results in Figure 5 are worth highlighting?

We edited the entire paragraph accordingly and moved in to the summary section (L664-680).

"Both simulations" should say "The two simulations".

Changed "between both simulations" to "between the two simulations" throughout the manuscript.

L402: "as well as current transport" – I think seasonal variations in current data (figs 4 and 5) have only been presented at 23W and 35W ?

We edited the entire paragraph accordingly and moved in to the summary section (L664-680).

L 414: "significant trends". Section 3.3. on interannual variations talks about significant trends when most of them seem to be related to natural variability and the differences between the wind forcings are very large compared with the variation in the JRA wind anomalies. You need to explain in the

methods section how significance is being assessed (and perhaps comment on what it means). In general I'm afraid I don't find the results in this section very convincing.

Apologies for not being clear enough. This section is mainly summarizing the results of Brandt et al. (2021) and we repeated calculations in JRA as an introduction to longer term variability (which appear as trend in the shorter observational time series). However, we realise that we did not differentiate enough between what have been previously found and what is part of our study which leads to the impression described by the reviewer. We edited the paragraph for more clarity. (L477-508)

L428: It is very difficult to assess the reliability of decadal variations in the re-analysis wind products as the observation system on which they are based fluctuates and the number of decades that have been observed is very limited.

Thank you for highlighting. We added a few sentences at the beginning of the section to highlight this. (L502-505)

L437: I think these volume transports (Figure 10) are again defined using Eq (2) and Table 1. This should be made completely clear as we are being asked to compare with transport defined over the full depth in fig 11.

Thank you for highlighting. We added a sentence at the beginning of the section. (L506-507)

Fig 9 caption: The penultimate sentence should be included in the description of Fig 1 or in the text discussing Fig 1.

Thank you for highlighting. We added the sentence to the caption of Fig. 1.

L447: anomalous -> anomalously ; do you mean compared to observations?

Changed. (L548)

L474: "before 1980" but not before 1970 in CORE?

The switch from positive to negative anomalies can be interpreted as a decrease of NEUC transport. We edited the sentence for clarity. (L561)

L480: "west of 20W" – do you mean east of 20W?

Thank you for highlighting. We edited the paragraph for clarity. (L556-L575)

L481: What longitude range do you mean when you mention the western NEUC (e.g. west of 20W)?

West of 30°W. Edited the paragraph for clarity. (L556-L575)

L488: "wind stress curl south". This does not seem to be a typo for "north". I find the statement quite puzzling.

Thank you for highlighting. This statement is wrong. Burmeister et al. (2019) found that the wind stress curl in TRATL01 can explain up to 40% of NEUC variability on interannual timescale (with a 2-5-years bandpass filter applied to time series). They found maximum positive correlation (R = 0.6) in the eastern basin of the tropical North Atlantic between 2°N and 8°N (Guinea Dome region). We revised the entire paragraph (L576-594)

**Summary**

L559-560: If this is the case, why don't these wind stress differences also drive differences in the currents at 35oW (i.e. further to the west)?

Thank you for highlighting that the summary did not differentiate enough how differences in the wind field impact the different current transport. We edited the entire summary for clarity and completeness.

L 561-568: This para and Fig A5 is an important part of the discussion but it seems strange to discuss it for the first time in the summary. Could it not be discussed immediately after the discussion of Fig 1. Perhaps Fig 1 could be adjusted to contain Fig A5.

Thank you for highlighting. As mentioned above we added a paragraph in the results section. (L360-367)

L 576-577: "As the subsurface off-equatorial …" The wind stress driving of the EUC is not much more direct is it? The sentence would be easier to read if it ended "than the EUC" (particularly as the EUC has not been discussed in the previous sentence or two).

We are afraid that we do not fully understand what the reviewer is referring to. The EUC is mainly driven by the easterly winds along the equator. Persistent easterly winds along the equator push the surface waters towards the west causing the thermocline to slope upwards to the east and hence driving the eastward flowing EUC in the subsurface along the equator (Wacongne, 1989). Other processes contributing to EUC variability are Ekman divergence (e.g. Brand et al., 2021) or wind stress curl between 2°S and 2°N (Assene et al, 2006) or resonant Equatorial basin modes which are again wind-driven. All these processes are wind-driven. The driving mechanism for the strength and variability of the off-equatorial undercurrents (SEUC, NEUC) are not fully understood and several theories exist (see Assene et al. (2020) for a nice summary). Assene et al. (2020) investigated the formation and maintenance of the off-equatorial subsurface currents in the Gulf of Guinea and highlighted the link between sub-mesoscale processes, mesoscale vortices and mean currents which can include any of the driving mechanisms suggested in previous studies: eddy fluxes (Jochum and Malanotte-Rizzoli,2004), meridional advection (Wang, 2005; Johnson and Moore, 1997; Marin et al., 2000; Hua et al., 2003; Marin et al., 2003; Ishida et al., 2005), lateral diffusion of vorticity (McPhaden, 1984) and the pull by upwelling in the eastern basin (McCreary et al., 2002; Furue et al., 2007, 2009).

L 581: "contribute" as mentioned earlier this could also be "partly due to"

Main instabilities causing tropical instability waves arise from barotropic instability of the EUC/nSEC, the nSEC/NECC and baroclinic instability within the nSEC and cSEC. We do not think that differences in the SEUC and NEUC may cause discrepancies in the appearance of the tropical instability waves. We added a sentence to explain this (L463-465).

L 588-589: "Most of the differences …" This is an important conclusion. I'm not really convinced that

it is fully justified.

Thank you for highlighting. We edited the paragraph to tone down our statement. (L681-700)

**Conclusions**

I'm not sure how the conclusions and the summary sections differ. Is the division really needed?

Thank you for this suggestion. We merged the two sections.

L 601-602: "The model results support …" This is also an important conclusion. But it is difficult to be sure about multidecadal variability from such a short period. Perhaps there could be some discussion here of what simulations would be needed to obtain firm conclusions (e.g. 500 year simulations?)

Thank you for highlighting. We acknowledge that the model simulations are too short to investigate variability on multidecadal timescales and edit the text throughout the manuscript to only refer to decadal variability. Furthermore, we included a sentence on that one would need several 100-yr long integrations to make sound statements about multi-decadal variability (L711-712).

L 606: It is strange to refer to Fig A4 for the first time here.

Thank you for highlighting. We included the reference to Fig. A4 in the section about NECC longer term variability where it should have been included previously. (L617)

Ascani, F., Firing, E., Dutrieux, P., McCreary, J. P. and Ishida, A.: Deep Equatorial Ocean Circulation Induced by a Forced–Dissipated Yanai Beam, Journal of Physical Oceanography, 40, 1118-1142, https://doi.org/10.1175/2010JPO4356.1, 2010.

Small, R. J., Curchitser, E., Hedstrom, K., Kauffman, B., and Large, W. G.: The Benguela Upwelling System: Quantifying the Sensitivity to Resolution and Coastal Wind Representation in a Global Climate Model*, Journal of Climate, 28, 9409–9432, https://doi.org/10.1175/JCLI-D-15-0192.1, 2015.

---

## Author Comment (AC2)

**Response to the reviews for**

**"Dependency of simulated tropical Atlantic current variability on the wind forcing"**

**by Kristin Burmeister, Franziska U. Schwarzkopf, Willi Rath, Arne Biastoch, Peter Brandt, Joke F. Lübbecke, and Mark Inall**

We thank all reviewers for the constructive comments. We edited the text accordingly. Additionally, we changed the colour of the ship sections in Fig.1 and added labels to Figure 8. Figure A1 in the appendix is new and was added to assess the applicability of the Sverdrup balance in the tropical Atlantic as a result of main comment three from Reviewer 2.

Detailed responses are found below. The comments by the reviewers are shown in black, our responses are given in red.

Review of "Dependency of simulated tropical Atlantic current variability on the wind forcing" submitted for Ocean Science by Kristin Burmeister, Franziska U. Schwarzkopf, Willi Rath, Arne Biastoch, Peter Brandt, Joke F. Lübbecke, and Mark Inall (egusphere-2023-1433)

This paper investigates wind-driven variability of upper layer currents in the tropical Atlantic Ocean by comparing two simulations with a high-resolution ocean circulation model forced by two different atmospheric products provided for ocean-climate modeling. Long-term mean states are compared first, then seasonal cycle and long-term variability including trends are compared. Observational results are also used to validate the model where possible.

Results presented would be very useful for enhancing our comprehensive understanding of mean state and variability of upper layer currents in the tropical Atlantic Ocean. Methods used are appropriate, though they may not be necessarily novel. Publication would be recommended after some revisions.

Major points.

It would be very helpful for readers if the key drivers, all of which should not be necessarily wind-driven, for currents discussed in the paper are summarized in the early part of the manuscript. For example, the easterly wind, the Ekman divergence, and the meridional gradient of Sverdrup stream function for EUC.

Thank you for this suggestion. We added a few sentences to better highlight links between the easterly winds along the equator and the EUC and the equatorial Ekman divergence and the zonal flow field (L36-37, L41-45, L63-65, L368-383). We also added a few sentences about the link between the difference of the Sverdrup stream function at different latitudes and the zonal flow in the Data and Method section under "Sverdrup Balance" (L260-265).

Specific points.

L7,69,366 etc.: It would be helpful where "the upwelling regions of the eastern tropical North Atlantic", which is referred to many times in the manuscript, specifically represent is indicated somewhere in the manuscript. Figure 1c would be one of candidates.

Thank you for this suggestion. Figure 1 is already very busy. Instead, we added a sentence to explain where the upwelling regions are located in the tropical Atlantic (L40-41).

L43: Does the Ekman transport diverge or converge south of the ITCZ?

Thank you for catching this. Ekman convergence occurs just south of the ITCZ and divergence occurs north of the ITCZ. Under the ITCZ there is the eastward flowing geostrophic NECC. We edited the text accordingly (L55-58).

L52–53: It would be worth mentioning that there are some resemblances of the structure of the zonal currents between the equatorial Atlantic and Pacific, as authors cited many papers that treated the tropical Pacific Ocean later in this paragraph.

Thank you for this suggestion. We added a sentence to the paragraph accordingly (L74-76).

L92: This is a nice approach, but it would be appropriate to describe specifically how the time gap was filled between the CORE simulation and observations somewhere in the manuscript.

We are sorry that this sentence caused confusion. What we meant was that the simulation forced by JRA55-do allows for a direct comparison between model and observations which is not possible for simulations forced by CORE. We edited the sentence accordingly (L110-111).

L179: To what is the EUC transport reconstruction using the mooring compared?

It is compared to the transport derived from the ship sections. We edited the sentence for clarity (L205-207).

L204: Are there any known reason for the differences found among phase speeds of the first baroclinic mode?

Thank you for highlighting. We found an error in our calculation. The model output was not interpolated to a sufficient vertical resolution before calculating $N^2$ which led to an erroneous mode decomposition. We corrected the text (L231-232) and table 2 accordingly (here table 1).

*Table 1: Result for gravity waves speeds in m/s for first five baroclinic modes of the gravest basin mode derived from the corrected $N^2$ profiles. Model and observations agree well with each other for all modes.*

|        | Mode 1 | Mode 2 | Mode 3 | Mode 4 | Mode 5 |
|--------|--------|--------|--------|--------|--------|
| Obs    | 2.51   | 1.40   | 0.98   | 0.76   | 0.57   |
| JRA    | 2.53   | 1.43   | 1.05   | 0.81   | 0.58   |
| CORE   | 2.51   | 1.42   | 1.04   | 0.80   | 0.57   |

L365: It would be nice to show that the JRA55 wind forcing actually captures fine structures that force NEUC variability.

Sorry if we did not articulate us clearly. JRA55 resolves fine wind stress curl features which are not present in CORE which is shown in Fig. 3. CORE shows large-scale high amplitudes in the annual cycle of the wind stress curl above the eastern upwelling regions (Fig. 3f) which are not present in JRA. The latter shows much finer local structures (Fig. 3 e). We added a reference to the figure accordingly (L437).

L406–408, 577–578: Are only first two baroclinic modes used for the harmonic analysis shown in Figures 4 and 5? If so, please be specific about this.

In Figure 4 and 5 we only fit harmonic annual and semi-annual cycles to the data. No mode decomposition is performed here. These are presented in Fig. 6 and Fig. A2.

L412: Since long-term trends are treated here, it would be good to discuss why trends are relevant or whether any issues have been raised in relation to global warming.

Thank you for the suggestion. We added two sentences in the text (L 478-479).

L543: "This agrees with the weakening easterly winds along the equator." … What does "this" point to? Should "weakening" be replaced with "trend of"? Or should "in JRA" be appended at the end of this sentence?

Thank you for highlighting that this section was not easy to follow. We edited the entire paragraph for clarity. (L635-639)

Technical points.

L62,63: Please give definition for NBUC and NBC.

Added. (L80-83)

L95: It would be more appropriate to cite Large and Yeager (2009) for introducing the CORE v2 dataset.

Thank you for highlighting. We edited the reference accordingly. (L146)

L121: well-establish ---> well-established

Changed. (L143)

L124: relative ---> relatively

Changed. (L147)

L129,757: It would be appropriate to cite Kobayashi et al. (2015; JMSJ) for introducing JRA-55.

Thank you for highlighting. We edited the reference accordingly. (L153)

L136: I would suggest something like, "Meridional ship sections are taken between 21°W and 28°W for velocity (31 sections) and hydrography and oxygen (22 sections) between 2000 and 2018".

Thank you for this suggestion. We edited the text accordingly. (L159-163)

Table 1, L157: It would be needed to indicate that $Z_u$ and $Z_l$ are determined by the depths of specific values of potential density.

Thank you for pointing this out. We edited the text and caption of Table 1 accordingly.

L246: establish ---> established

Changed. (L280)

L274: extend ---> extent

Changed. (L310)

L275: reasonable ---> reasonably

Changed. (L312)

L276: The transports of currents north of the equator from the two simulations diverge…

Changed. (L313-314)

L296: 2°S and 2°S ---> 2°S and 2°N?

Changed. (L370)

L344–345: Are only eastward components integrated? If so, please notify it.

Yes, we integrate over eastward velocities as stated in L414-416: "To compare transport from model output and moored observations at 23°W, we calculated the transport for the EUC and NEUC from model output as integral of eastward velocities in the respective box" as well as in the methodology section.

L345: 1°12'S-1°12'N

 Changed. (L416)

L345: Furthermore

Changed. (L416)

L376: force ---> forced

Changed. (L449)

L389: created ---> create

Changed. (L462)

L421: extent ---> extend

The whole paragraph was edited for clarity. (L477-508)

L439–440: It would be helpful if what is done here is explained using equations (e.g., $U = \Psi_N - \Psi_S$).

Thank you for this suggestion. We added a formula in the Data and Method section (L260-265) and referenced it in the text accordingly (L348).

L447,481: anomalous ---> anomalously

Changed. (L545)

L476: we find significant negative trends in current transport east of ~30°W in JRA$_{sim}$.

Changed. (L564)

L480: west of ---> east of (?)

Thanks for highlighting. Edited paragraph for clarity. (L566-L584)

L526–527: despite the distinct inter-simulation discrepancies of the NECC on multidecadal timescales, especially after year 2000.

Changed. (L617-618)

L564: wester ---> western

Changed. (L631)

L573: asses ---> assess

Changed to "To investigate…". (L667)

L605: strength the ---> strength and the

Changed. (L715)

Figure 2 (the last line of the cation): as well as

Changed.

Figure 3: Is the phase given in month of the year when zonal wind stress/wind stress curl is maximum?

Yes, we added a sentence to the caption.

Figure 8: Please add labels *a* through *l* to panels.

Changed.

Figure 11 (caption, the line above the bottom): different ---> difference

Changed.

Figure A5: wind stress curl (above) and Sverdrup transport (below)

Changed.

L784: Perez et al. (2021) … Remove abstract.

Changed.

Large, W. G. and Yeager, S. G.: The global climatology of an interannually varying air-sea flux data set, Clim. Dynam., 33, 341–364, https://doi.org/10.1007/s00382-008-0441-3, 2009.

Kobayashi, S., Ota, Y., Harada, Y., Ebita, A., Moriya, M., Onoda, H., Onogi, K., Kamahori, H., Kobayashi, C., Endo, H., Miyaoka, K., and Takahashi, K.: The JRA-55 reanalysis: general specifications and basic characteristics, J. Meteorol. Soc. Jpn., 93, 5–48, https://doi.org/10.2151/jmsj.2015-001, 2015.

Ascani, F., Firing, E., Dutrieux, P., McCreary, J. P. and Ishida, A.: Deep Equatorial Ocean Circulation Induced by a Forced–Dissipated Yanai Beam, Journal of Physical Oceanography, 40, 1118-1142, https://doi.org/10.1175/2010JPO4356.1, 2010.

Small, R. J., Curchitser, E., Hedstrom, K., Kauffman, B., and Large, W. G.: The Benguela Upwelling System: Quantifying the Sensitivity to Resolution and Coastal Wind Representation in a Global Climate Model*, Journal of Climate, 28, 9409–9432, https://doi.org/10.1175/JCLI-D-15-0192.1, 2015.

---

## Referee Report (RR1)

**Dependency of simulated tropical Atlantic current variability on the wind forcing**

**submitted to Ocean Sciences**

The paper has been very substantially revised. It has been significantly improved in many ways (e..g the abstract and summary are better; much of the discussion of results is more quantitative and precise; the qualifications/uncertainties are summarised; there are some interesting additional results ). Nearly all the responses to my suggestions are  satisfactory. My opinion is that the paper should be accepted for publication following some relatively minor further revisions.

Given the number of substantial changes made, it's not surprising that some further iteration is likely to be worthwhile. I've made a number of suggestions below, nearly all of them quite minor, which could help to improve the paper's presentation. I start by commenting on the response to one of my main comments on the earlier draft.

**Responses to my comments**

Thank you for taking the trouble to include as part of your responses some additional figures. From the additional figure 1, I can see that the Ekman pumping fields are more or less what one might naively expect from those for the wind stress curl. The meridional Ekman transport fields for CORE and JRA are interesting (and at present the time-mean zonal wind fields which give rather similar information are not displayed in any of the figures) though again the differences between them, in the right hand plot, don't look large. As the various fields play different roles in the dynamics I still think there is a case for including more of those of figure 1 within one of the figures of the paper. But it is quite late in the day to do that, to get the most out of the figures showing the differences between fields one would probably need to use a more discerning scale, and the discussion now mentions these fields in an appropriate way. So I leave this to the authors' discretion.

**Main suggestions**

There are some new results quoted in the abstract about interannual to decadal variations in the nSEC and NECC. These seem to be genuinely interesting results but they are not presented until the very end of the results section (subsections 3.3.5 and 3.3.6), only one of them is illustrated by a figure, and that figure is in the appendix. Could a figure illustrating these results be included in the main body of the paper? Might sub-sections 3.3.5 and 3.3.6 come earlier in section 3.3?

**Additional suggestions**

1. Some aspects of the wording could be improved. In particular:
    - the word "both" still appears 58 times. In many (but not all) cases it should be replaced by "the two" (e.g. lines 6 and 16)
    - the subject and the verb need to agree (both should either be singular or plural) (e.g. line 8)
    - the tense of the verbs should be checked –  the past tense is over-used

    Perhaps Mark Inall as a native English speaker could review these aspects of the text.

2. Line 20: commonplace is one word. I can see that this sentence is trying to say something important but in my view it still doesn't quite work.
3. Line 27: ecosystem should be plural here

4. Lines 63-64. Later you say that the Sverdrup transport also influences the EUC (which agrees with the description of the EUC in Vallis' textbook 2017 section 22.3) . I suppose you are not implying here that the wind on the equator is the only factor driving the EUC but it could be read that way.

5. Figure 1: Are the tiny black arrows on figures 1a, 1b and 1c surface wind stress vectors? I don't see any mention of them in the text or figure caption. If you keep the arrows would it be possible to make them slightly larger (at least in figure 1c) ?

6. Line 104: replace "having additionally" by "with"

7. Line 124: described -> describe

8. Line 306: do you mean the near-equatorial currents? Geostrophy of course does not hold at the equator.

9. Line 309: change "this section" to "the 23oW section" and delete "along 23oW section" later in sentence

10. Line 338: The Angola Dome region is rather narrow in longitude; is the Ekman pumping of the SEUC really confined to this region? Also my impression from Fig 1 is that this region is somewhat south of the core of the SEUC.

11. Line 348: $U_\Psi$ is the zonal transport between the N and S "bounds". It is strange to call it "the meridional divergence of the meridional Sverdrup flow"

12. Line 409 – incoherence isn't the right word

13. Line 469: "boundary conditions" -> "parameters"

14. Line 482 is -> it

15. Line 519: delete "meridional"

16. Figure 8 caption: "annual mean zonal wind stress anomalies with respect to the seasonal cycle" – is that what you mean to say?

17. Figure 9 caption (b-g) should be (b-f)

18. Line 555: "The transitioned" is "AMV" missing ?

19. Lines 572-573: This sounds quite a significant result (R=0.75). It would be nice to see a figure illustrating it (somewhat similar to Fig A5) so that the reader can better judge its significance.

20. Line 583: delete second "on"

21. Figure 10: The labels for CORE and JRA are really very small

22. Line 661: "meridional Sverdrup" I think should be "zonal Sverdrup"

23. Line 617 & Figure A5: This is one of rather few results that is highlighted in the abstract. Figure A5 looks very convincing. Shouldn't it be one of the figures in the main text rather than the appendix? (see main suggestion above)

24. Line 623-624: This result is highlighted in the abstract. It would be good if it could be illustrated in a figure.

25. Line 638-639: "However…" please check this sentence.

26. Line 648: Figs 2 and 9 do not show wind stress fields?

27. Line 651: please check the position of "across the entire basin" in this sentence

28. Line 665: "Both" -> "The two"

29. Line 669: "are" -> "can be"

30. Line 673: "a strong" -> "an overly strong"

31. Line 675:  I find this difficult to see. Was it highlighted earlier?

32. Line 684: omit "meridional"

33. Line 698-700: refer to Fig A5?

34. Line 719: "both" -> "the two"

35. Line 734: "onto" -> "on"

---

## Author Response (AR2)

**Response to the reviews for revised version of**

**"Dependency of simulated tropical Atlantic current variability on the wind forcing"**

**by Kristin Burmeister, Franziska U. Schwarzkopf, Willi Rath, Arne Biastoch, Peter Brandt, Joke F. Lübbecke, and Mark Inall**

We thank all reviewers for the constructive comments. We edited the text accordingly. Additionally, we increased the size of the wind stress arrows in Figure 1, increased the label font size in Figure 10 and moved former Figure A5 into the main manuscript (now Figure 12) extending it with timeseries for the NEUC and nSECu as suggested by the reviewer.

Detailed responses are found below. The comments by the reviewers are shown in black, our responses are given in red.

Review (by reviewer 2) of **Burmeister et al.**

**Dependency of simulated tropical Atlantic current variability on the wind forcing**

**submitted to Ocean Sciences**

The paper has been very substantially revised. It has been significantly improved in many ways (e.g. the abstract and summary are be er; much of the discussion of results is more quantitative and precise; the qualifications/uncertainties are summarised; there are some interesting additional results). Nearly all the responses to my suggestions are satisfactory. My opinion is that the paper should be accepted for publication following some relatively minor further revisions.

Given the number of substantial changes made, it's not surprising that some further iteration is likely to be worthwhile. I've made a number of suggestions below, nearly all of them quite minor, which could help to improve the paper's presentation. I start by commenting on the response to one of my main comments on the earlier draft.

Responses to my comments

Thank you for taking the trouble to include as part of your responses some additional figures. From the additional figure 1, I can see that the Ekman pumping fields are more or less what one might naively expect from those for the wind stress curl. The meridional Ekman transport fields for CORE and JRA are interesting (and at present the time-mean zonal wind fields which give rather similar information are not displayed in any of the figures) though again the differences between them, in the right hand plot, don't look large. As the various fields play different roles in the dynamics I still think there is a case for including more of those of figure 1 within one of the figures of the paper. But it is quite late in the day to do that, to get the most out of the figures showing the differences between fields one would probably need to use a more

discerning scale, and the discussion now mentions these fields in an appropriate way. So I leave this to the authors' discretion.

Thank you for the feedback. Since we feel that the additional plots do not add substantial further information and the paper contains already 12 figures we decided against including another figure.

Main suggestions

There are some new results quoted in the abstract about interannual to decadal variations in the nSEC and NECC. These seem to be genuinely interesting results but they are not presented until the very end of the results section (subsections 3.3.5 and 3.3.6), only one of them is illustrated by a figure, and that figure is in the appendix. Could a figure illustrating these results be included in the main body of the paper? Might sub-sections 3.3.5 and 3.3.6 come earlier in section 3.3?

Thank you for this suggestion. We extended former Figure A5 with a comparison between zonal wind stress and the NEUC as well as the nSECu and included it in the main text (now Figure 12). For consistency, the order of currents the sections and figures is generally the same (EUC, NEUC, SEUC, NECC, nSEC). This is why the results for the NECC and nSEC are coming last. To be consistent, we decided not to change the order of section 3.3.

Additional suggestions

1. Some aspects of the wording could be improved. In particular:

- the word "both" still appears 58 times. In many (but not all) cases it should be replaced by "the two" (e.g. lines 6 and 16) –

  Changed throughout the manuscript.

- the subject and the verb need to agree (both should either be singular or plural) (e.g. line 8)

  Changed throughout the manuscript.

- the tense of the verbs should be checked – the past tense is over-used

  Changed throughout the manuscript.

Perhaps Mark Inall as a native English speaker could review these aspects of the text.

2. Line 20: commonplace is one word. I can see that this sentence is trying to say something important but in my view it still doesn't quite work.

Thank you for pointing this out. We shorten the sentence for clarity (L20-22).

3. Line 27: ecosystem should be plural here

Changed (L 27).

4. Lines 63-64. Later you say that the Sverdrup transport also influences the EUC (which agrees

with the description of the EUC in Vallis' textbook 2017 section 22.3) . I suppose you are not implying here that the wind on the equator is the only factor driving the EUC but it could be read that way.

Thank you for pointing this out. We edited the sentence for clarity (L63-65)

5. Figure 1: Are the tiny black arrows on figures 1a, 1b and 1c surface wind stress vectors? I don't see any mention of them in the text or figure caption. If you keep the arrows would it be possible to make them slightly larger (at least in figure 1c) ?

Thank you for pointing this out. We edited the caption and increased the size of the arrows for clarity.

6. Line 104: replace "having additionally" by "with"

Changed (L105).

7. Line 124: described -> describe

Changed (L114).

8. Line 306: do you mean the near-equatorial currents? Geostrophy of course does not hold at the equator.

Thank you for highlighting. We edited the sentence accordingly (L307).

9. Line 309: change "this section" to "the 23oW section" and delete "along 23oW section" later in sentence

Changed (L309).

10. Line 338: The Angola Dome region is rather narrow in longitude; is the Ekman pumping of the SEUC really confined to this region? Also my impression from Fig 1 is that this region is somewhat south of the core of the SEUC.

The region between 7.5° and 4.5°S, 0.5°W and 2.5°E forms a subregion within the entire Angola Dome region. Doi et al. (2007, doi: 10.1175/2007JPO3552.1) found that two domes exist within the Angola dome region, with a weaker dome centred at 6°S, 1°E. The authors found the upwelling of the weaker dome to be associated with changes in the SEUC strength. The core position of the SEUC varies between 4°-5.5°S overlaps most of the time with the northern boundary of the weak dome region. Similarly, the core of the NEUC is located just south of the Guinea Dome region.

11. Line 348: $U_\psi$ is the zonal transport between the N and S "bounds". It is strange to call it "the meridional divergence of the meridional Sverdrup flow"

Thank you for highlighting. We edited the sentence accordingly (L348-349).

12. Line 409 – incoherence isn't the right word

Thank you for pointing this out. We edited the sentence accordingly (L410).

13. Line 469: "boundary conditions" -> "parameters"

Changed (L424)

14. Line 482 is -> it

Changed (L483).

15. Line 519: delete "meridional"

Deleted (L520).

16. Figure 8 caption: "annual mean zonal wind stress anomalies with respect to the seasonal cycle" – is that what you mean to say?

Yes. The anomalies are calculated by removing the seasonal cycle (1980-2009) from the monthly mean output before temporally averaging to annual resolution. We edited the caption for clarity.

17. Figure 9 caption (b-g) should be (b-f)

Changed.

18. Line 555: "The transitioned" is "AMV" missing ?

Added (L556).

19. Lines 572-573: This sounds quite a significant result (R=0.75). It would be nice to see a figure illustrating it (somewhat similar to Fig A5) so that the reader can be er judge its significance.

Thank you for this suggestion. We extended former Figure A5 with a comparison between zonal wind stress and the NEUC as well as the nSECu and included it in the main text (Figure 12).

20. Line 583: delete second "on"

Deleted (L513).

21. Figure 10: The labels for CORE and JRA are really very small

Thank you for highlighting. We increased the font size of all labels in Fig. 10.

22. Line 661: "meridional Sverdrup" I think should be "zonal Sverdrup"

Changed (L662).

23. Line 617 & Figure A5: This is one of rather few results that is highlighted in the abstract. Figure A5 looks very convincing. Shouldn't it be one of the figures in the main text rather than the appendix? (see main suggestion above)

Please see answer to main suggestions above.

24. Line 623-624: This result is highlighted in the abstract. It would be good if it could be illustrated in a figure.

Please see answer to main suggestions above.

25. Line 638-639: "However…" please check this sentence.

Thank you for highlighting that the sentence was not clear. We edited it for clarity (L639-642).

26. Line 648: Figs 2 and 9 do not show wind stress fields?

Thank you for pointing out that we referenced to the wrong figures. We corrected that (L652).

27. Line 651: please check the position of "across the entire basin" in this sentence

Thank you for highlighting. We edited the sentence for clarity (L654-655).

28. Line 665: "Both" -> "The two"

Changed (L668).

29. Line 669: "are" -> "can be"

Changed (L672).

30. Line 673: "a strong" -> "an overly strong"

Changed (L676).

31. Line 675: I find this difficult to see. Was it highlighted earlier?

32. Line 684: omit "meridional"

Changed (L687).

33. Line 698-700: refer to Fig A5?

Changed (L702).

34. Line 719: "both" -> "the two"

Deleted both (L722).

35. Line 734: "onto" -> "on"

Changed (L644).